



# Improving prediction of trans-boundary biomass burning plume dispersion: from northern peninsular Southeast Asia to downwind western north Pacific Ocean

Maggie C. Ooi[1,2], Ming-Tung Chuang[3], Joshua S. Fu[4], Steven S. Kong[1], Wei-Syun Huang[5], Sheng-Hsiang Wang[1,5], Andy Chan[6], Shantanu K. Pani[1], Neng-Huei Lin[1,5]

[1] Department of Atmospheric Sciences, National Central University, Taoyuan, 32001, Taiwan
[2] Institute of Climate Change, National University of Malaysia, Bangi 43600, Malaysia
[3] Research Center for Environmental Change, Academia Sinica, Taipei, 11529, Taiwan
[4] Department of Civil and Environmental Engineering, University of Tennessee, Knoxville, 37996, USA
[5] Center for Environmental Monitoring Technology, National Central University, Taoyuan, 32001, Taiwan
[6] Department of Civil Engineering, University of Nottingham Malaysia, Semenyih, 43500, Malaysia.

*Correspondence to*: Neng-Huei Lin (nhlin@cc.ncu.edu.tw)

**Abstract.** The boreal spring biomass burning (BB) in the northern peninsular Southeast Asia (nPSEA) are lifted into the subtropical jet stream, get transported and deposited across nPSEA, South China, Taiwan, and even the western North Pacific Ocean. This paper as part of the 7-Southeast Asian Studies (7-SEAS) project effort attempts to improve the prediction capability of the chemical transport model (WRF-CMAQ) over a vast region including the mountainous near-source burning sites at nPSEA to its downwind region. Several sensitivity analyses of plume rise are compared in the paper and it discovers that the initial vertical allocation profile of BB plume and plume rise module (PLMRIM) are the main reasons causing the inaccuracies of the WRF-CMAQ simulations. The smoldering emission from the Western Regional Air Partnership (WRAP) empirical algorithm included has improve the accuracies of $PM_{10}$, $O_3$ and CO at the source. The best performance at the downwind sites is achieved with the inline PLMRIM that accounts for the atmospheric stratification at the mountainous source region with the high-resolution FINN burning emission dataset. The calibrated model greatly improves not only the BB emission prediction over near-source and receptor ground-based measurement sites but also the aerosol vertical distribution (MPLNET, CALIPSO) and column aerosol optical depth (MODIS AOD) of the BB aerosol along the transport route. Three distinct transport mechanisms from nPSEA to the western North Pacific are then identified while a particular mechanism which involves Asian cold surge is able to mix the BB smoke plumes into the boundary layer and affects the ground surface over the western Taiwan.

## 1 Introduction

Large amounts of gaseous and aerosol pollutants released from biomass burnt affect regional air quality, radiative forcing, public health, and economic burden, especially in Southeast Asia (Chen et al., 2017; Lee et al., 2017; Pani et al., 2018,





2020). The prolonged heat during the dry season (December to May) in peninsular Southeast Asia (PSEA) has led to the deterioration of biomass burning (BB) in northern PSEA (nPSEA) (Kim Oanh and Leelasakultum, 2011). The outflow of the BB smoke plumes from nPSEA usually occurs during the spring season (late-February until mid-April) when the high-pressure system has retreated northwards back into the Asian Continent. The mountainous structure over the northcentral PSEA has lifted the BB plume into the subtropical Pacific High (700 to 800 hPa, ~1-3 km) under prevailing south wind (Dong and Fu, 2015b; Huang et al., 2020). The plume is then transported eastward to the West Pacific and frequently detected at the Lulin Atmospheric Background Station (LABS) in central Taiwan (Fu et al., 2012; Lee et al., 2011; Lin et al., 2017, 2014, 2013; Ou-Yang et al., 2014; Wang et al., 2013b). Moreover, there were several instances when the high-pressure system entered Taiwan and brought the upper-layer BB plumes down to populous southwestern Taiwan and altered the atmospheric chemistry and composition (Dong et al., 2018; Huang et al., 2016; Yen et al., 2013).

Space-borne remote-sensing data from satellites and the high spatiotemporal data generated from the chemical weather prediction (CWP) model are often used for studying long-range transport of BB smoke across the region (e.g. Huang et al., 2020; Tsay et al., 2013). Previous studies have found that the numerical model has prone to overpredict the BB emissions including CO, $PM_{2.5}$, and $PM_{10}$ up to three times of the measured amount at the major burning source in northern Thailand (Huang et al., 2013; Pimonsree et al., 2018). The exceedance of predicted emission at the near-source burning leads to the incorrect modelled signal at the downwind site (Fu et al., 2012). The modelled columnar aerosol optical depth (AOD) are found comparable with aerosol products of Aerosol Robotic Network (AERONET) and Moderate Resolution Imaging Spectroradiometer (MODIS) sensor as well as columnar CO and $NO_2$ at the burning source over nPSEA region but great discrepancies are found for the spatial distribution of downwind plumes (Dong and Fu, 2015b; Fu et al., 2012). In those models, the vertical distribution percentage of BB was set to be constant throughout the case. However, there are many possible factors that govern the actual plume rise condition, including the fire size, vegetation cover, buoyancy heat flux, wind drag, boundary layer condition, etc. (Freitas et al., 2010; Kukkonen et al., 2014; Paugam et al., 2016; Val Martin et al., 2012). Furthermore, the accuracy of the model depends greatly on the plume rise condition.

As part of the local effort of interdisciplinary 7-Southeast Asian Studies (7-SEAS) project (Lin et al., 2013; Reid et al., 2013), this paper attempts to improve the modelling performance of the long-range transport of BB from the nPSEA region to the downwind region using the WRF-CMAQ model. The paper attempts to improve the ability of the Community Multiscale Air Quality (CMAQ) model and its plume rise module (PLMRIM) to predict the complexity of BB amount from its burning source in nPSEA to its downwind receptor LABS. With the availability of on-site and satellite LiDAR (Light Detection and Ranging) measurement, the vertical plume rise profile can be better understood to ensure that BB plumes are distributed according to the actual conditions (Walter et al., 2016; Wang et al., 2013b). In this work, several factors including the injection height, initial vertical distribution, and smoldering fraction are considered into the model. Knowing that the atmospheric circulation over nPSEA is also affected by terrain, the work now intends to incorporate the interaction of the

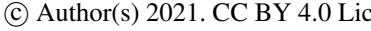
atmospheric stratification to the PLMRIM. This research approaches mainly from the perspective of the vertical distribution profile of modelled BB emission with the assistance of top-down and bottom-up vertical LiDAR profilers. The better-performing setting will be applied to test its applicability and to dissect the sources of high pollution at LABS and in western Taiwan.

The model experimental design (Section 2.1), model emission input (Section 2.2), and case study setup (Section 2.3) are explained in detail. The performance of the PLMRIM is then verified with ground-based measurement station in Section 3.1 and vertical aerosol products from LiDAR sensors (MPLNET, CALIPSO) and MODIS columnar AOD (Section 3.2), where the reliability and accuracy of inline PLMRIM are discussed (Section 3.3). The resulting output is subsequently studied in

Section 4 to answer the transport mechanism to the ground-based observation sites in western Taiwan. From which conclusion to the findings are made in Section 5.

## 2 Methodology

The study focuses on the spring BB events in March 2013. With moderate burning occurring in nPSEA, this ENSO-neutral year is chosen because the LABS mainly received the BB plumes with minimal influence from the Asian dust storm to

Taiwan (NOAA-ESRL, 2020; TAQM, updated daily; Kong et al., 2021 in review). The 7-SEAS spring campaigns carried out during the BB season supplies abundance of data to the near source burning and receptor.

### 2.1 Model Physics and Experimental Design

This work employs Weather Research and Forecast (WRF-ARW v3.9.1) (Wang et al., 2017) model to hindcast the weather field and predict the corresponding air chemistry field with the chemical transport model CMAQ v5.2.1 (Byun and Schere,

2006). The model domain is dynamically nested down from the majority of Asia (d01 resolution: 45 km) to cover the transport route from nPSEA to Taiwan (d02: 15km), Taiwan only (d03: 5km) and nPSEA only (d04: 5km) as shown in Fig. 1. The weather input for the initial and lateral boundary condition is the 6-hourly 1° x 1° National Centers for Environmental Prediction (NCEP) Final Analyses (FNL) dataset (NCEP-ds083.2, Updated daily). As an extension of the latter, data assimilation is applied for both grid- and observation-nudging. The weather data for observation nudging are obtained from

NCEP ADP Global Surface (NCEP-ds461.0, Updated daily) and Upper Air Observational Weather Data (NCEP-ds351.0, updated daily) with additional local sites operated by Taiwan Central Weather Bureau (CWB) and Thailand Pollution Control Department (PCD). The radii of influence (RIN) for both d03 and d04 are updated to 100 km based on the average distance between the observation stations (d03: 125 km, d04: 153 km) and the minimum distance between 2 stations (d03: 64 km, d04: 36 km). Wind speed and wind direction are substantially improved by observation nudging. A detailed discussion

about meteorology performance is given in Appendix A. Other WRF-CMAQ settings and configurations are listed in Table 1.





The Micro-Pulse Lidar Network (MPLNET) is a federated network managed by NASA to measure the aerosol vertical structure (Welton et al., 2000). In line with the 2014 7-SEAS spring campaign conducted in nPSEA, the gridded extinction, diagnosed from the planetary boundary layer height and vertical aerosol extinction coefficient data collected is used to verify the performance of the model output (Wang et al., 2015a). The top-down lidar system, the Cloud-Aerosol Lidar with Orthogonal Polarization (CALIOP) on the Cloud-Aerosol Lidar and Infrared Pathfinder Satellite Observations (CALIPSO) satellite is used to study the transport pattern over larger spatial coverage to complement the single point cross-extinction profile provided by the MPLNET system. The diagnosed vertical feature mask (VFM) product is used to distinguish the aerosol types with consideration of observed backscatter strength and depolarization (Winker et al., 2011).

**Table 1: WRF and CMAQ model settings**

|                                            | Settings                                                          |
| ------------------------------------------ | ----------------------------------------------------------------- |
| **Weather model**                          | WRF version 3.9.1                                                 |
| **Period**                                 | 1– 31 Mar 2013 (after spin up)                                    |
| **Boundary condition**                     | NCEP FNL lateral boundary condition                              |
| **Vertical**                               | 41 layers up to 50 hPa with 10 layers in the bottom 2km          |
| **Weather nudging**                        | Grid and observation nudging                                     |
| **Planetary boundary**                     | Asymmetric Convective Mechanism 2                               |
| **Surface and land surface model**         | Pleim-Xiu                                                         |
| **Longwave radiation**                     | RRTM scheme                                                       |
| **Shortwave radiation**                    | Goddard                                                           |
| **Microphysics scheme**                    | Goddard                                                           |
| **Cumulus scheme**                         | Kain-Fritsch (1) for d01, d02 only                              |
| **Chemistry transport model**              | CMAQ version 5.2.1                                                |
| **Gas-phase chemistry and aerosol mechanism** | CB05e51 + AE6 (with aqueous chemistry)                        |
| **Emission inventory**                     | d01, d02: MICS-ASIA 2010, biogenic emission from MEGANv2.1<br>d03: Taiwan local emission inventory (TEDS v8.1) |



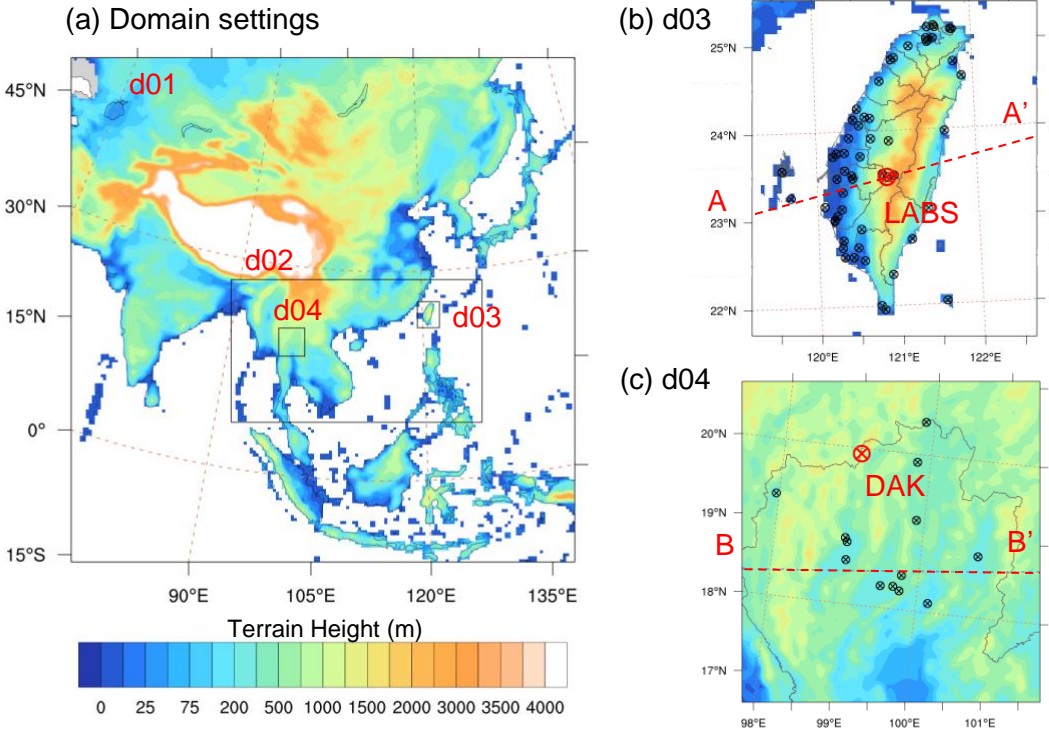

**Figure 1: (a) Domain setup of model (domain 1-4) with terrain height information; (b) 3rd domain covering Taiwan (d03) with information of terrain height (contour fill), AA' cross section (dotted red line), locations of Taiwan EPA air quality and CWB weather stations (black dots) and LABS receptor site (big red dot); (c) 4th domain covering part of nPSEA (d04) with terrain height (contour fill), BB' cross section (dotted red line), location of Thailand PCD ground air quality stations (black dots) and DAK source site (big red dot).**

## 2.2 Emission Data

### 2.2.1 Anthropogenic and biogenic emission inventories

The anthropogenic emissions are re-gridded for the 1st, 2nd and 4th domain (d01, d02, d04 in Fig. 1) from MIX dataset available at 0.25° x 0.25° for the year 2010 (Li et al., 2017; Zheng et al., 2018). Model of Emissions of Gases and Aerosols from Nature (MEGAN v2.10) produces the biogenic emission input (Guenther et al., 2012) using the updated 8-day averaged leaf area index (LAI) (Yuan et al., 2011) and present-day plant functional types (PFT) from the Community Land Model version 4.0 (CLM4.0) (Oleson et al., 2010). The 3rd domain (d03) covering Taiwan uses the 2010 anthropogenic and biogenic emissions from the locally developed Taiwan national emission database (TEDSv8.1) (TEPA, 2017). Except the high quality of the East Asia national emission inventories (China, Taiwan, Japan, and Korea), large uncertainties of Southeast Asia emission due to the scarce availability of region-specific emission factor are pointed out by the inventory developers (Kurokawa et al., 2013; Li et al., 2018; Ohara et al., 2007) and local modelling efforts (Dong and Fu, 2015a; Ooi et al., 2019). Such inaccuracies are likely to affect the performance of further modeling work in the area. Therefore, energy statistics based on global anthropogenic emissions dataset, Evaluating the Climate and Air Quality Impacts of Short-Lived





Pollutants (ECLIPSE) developed by International Energy Agency (IEA) (Klimont et al., 2017) is used in place of the MIX dataset for peninsular SEA (PSEA). The accuracy deviation between these two datasets in nPSEA is determined through the WRF-CMAQ model performance in Section 4. The detailed comparison of ECLIPSE and MIX dataset in 2010 is discussed in Appendix B.

### 2.2.2 Biomass burning emission inventory

The study region is composed of small fire while small area burnt but has a rather substantial amount of fuel load and BB emissions due to the high woody compositions of the tropical and temperate forest covers. The global data set, Fire INventory from NCAR (FINN v1.5) has been applied in several previous works of literature in the region (Lin et al., 2014; Pimonsree and Vongruang, 2018) and is used as the input to the BB emission inventory into the model. A particular comparison work done for 2014 biomass burning episodes has shown FINN when used with NCEP FNL boundary condition gives the greatest accuracy for $PM_{10}$ at the source region compared to the GFEDv4.1 fire emission dataset (Takami et al., 2020). Seeing that the temporal speciation is handled in this research work, the main difference between fire emission inventories is the total amount of emission produced (Liu et al., 2020), hence this paper will settle with regionally more robustly tested FINN dataset for the subsequent studies. FINN is a 1 km x 1 km resolution bottom-up daily emission dataset produced from the MODIS product of active fire, land-cover type, and vegetation continuous field (Wiedinmyer et al., 2011). Each active fire is assumed for a 1 $km^2$ burnt area and the emission factor is geographically and land-cover dependent. The BB emission is processed with the *fire_emis* preprocessor to allocate to each grid and specify to the hourly-scale for input into the WRF-CMAQ model.

### 2.3 Case study setup

The plume rise module (PLMRIM) derives the initial plume top and bottom, plume rise and its dispersion according to the atmospheric stability and its residual buoyancy flux (Kukkonen et al., 2014). Among a wide range of PLMRIM approaches, the simplest plume rise allocation method is the direct allocation of the initial plume top and bottom through prescribed height for all fires. This is the conventional method adopted in the case study region (Chuang et al., 2016b; Pimonsree et al., 2018). They can be determined on fixed height (Wang et al., 2013a), an empirical ratio of the plume height allocation (WRAP, 2004), adjusted with the stereo-height data from space-based Multi-angle Imaging Spectroradiometer (MISR) (Jian & Fu, 2014; Val Martin et al., 2012), etc. The inline plume rise algorithm couples the interaction of BB plumes dispersion with the basic weather dynamics to determine the effective plume rise height and subsequently the plume top and bottom. This inline PLMRIM is also able to resolve the fire on the sub-grid scale and feedback the plume dynamics information into the atmospheric dynamics (Gillani & Godowitch, 1999). However, the more complex the PLMRIM gets, the higher quality and quantity of input data are required to ensure its reliability.





In this work, combinations of injection height, initial vertical distribution, smoldering fraction, and offline and inline PLMRIM are tested to determine the more suitable settings for prediction of plume rise. Five case studies are set up for the evaluation of plume rise performance and their respective initial plume rise profiles are shown in Table 2. **Nofire** case

represents the pollution condition when no BB emission is included, while the others allocate the BB emission from the FINN dataset. **F800** and **F2000** represent the offline PLMRIM where the injection height is fixed at generally accepted 800 m and 2000 m (Wang et al., 2013a). This fixed height method controls the plume top to be consistent hence there is no hourly and daily variation of the plume top throughout the simulation period. **FWrp** uses the WRAP empirical equation to allocate the initial plume rise (WRAP, 2004). The plume top and bottom vary hourly with the buoyancy efficiency with

higher plume height during the hotter noontime as illustrated in the initial plume profile in Figure 2 (**FWrp**). However, the empirical ratio adopted for each burning grid is the same every day. **Idef** is the inline plume-in-grid system that comes with the CMAQ model (Gillani and Godowitch, 1999). Fire emission is fed into the model at each grid point with plume top and bottom calculated through interaction of plume buoyancy efficiency and atmospheric stratification. The vertical distribution of CO plume on 12 Mar 2013 is shown in Figure 2 (**Idef**), but the daily weather condition is expected to vary the vertical

distribution. **IWrp** has updated **Idef** with the WRAP empirical specification on burnt area size (also known as fire size). In this case, the plume can be distributed according to the diurnal buoyancy efficiency and near-surface smoldering fraction as specified by WRAP. With a more reasonable BB plume peak at the noontime in Figure 2 (**IWrp**), it is expected to improve the near-source concentration prediction of the model as seen from the initial plume profile. **IWrp+EC** is the same as **IWrp** but with the anthropogenic emission in PSEA replaced by the ECLIPSE dataset as specified in Section 3.2.1. The initial

emission profiles (within plume top and bottom) of all cases are distributed evenly according to the height of each vertical layer.



**Table 2: Case setup to evaluate PLMRIM performance**

| Fire emission | Plume rise module | Initial plume rise allocation (Injection height) | Time variant | Anthropogenic Emission (d01, d02, d04) |
|---|---|---|---|---|
| **Nofire** | | - | - | MIX |
| **F800** | No | *Plume top:* 0.8 km *Plume bottom:* 0 km *Smoldering fraction*: no | - | MIX |
| **F2000** | No | *Plume top:* 2.0 km *Plume bottom:* 0 km *Smoldering fraction*: no | - | MIX |
| **FWrp** | No | *Plume top and bottom & Smoldering fraction:* Fire heat flux and prescribed bins of acres burnt | Daily fire size | MIX |
| **IDef** | Inline | *Plume top and bottom*: 1.5 x effective plume rise height *Smoldering fraction*: yes | Daily atmospheric stability | MIX |
| **IWrp** | Inline | *Plume top and bottom*: 1.5 x effective plume rise height *Smoldering fraction*: FWrp | Daily fire size and daily atmospheric stability | MIX |
| **IWrp+EC** | Inline | Same as IWrp | Same as IWrp | Updated SEA region with ECLIPSE |

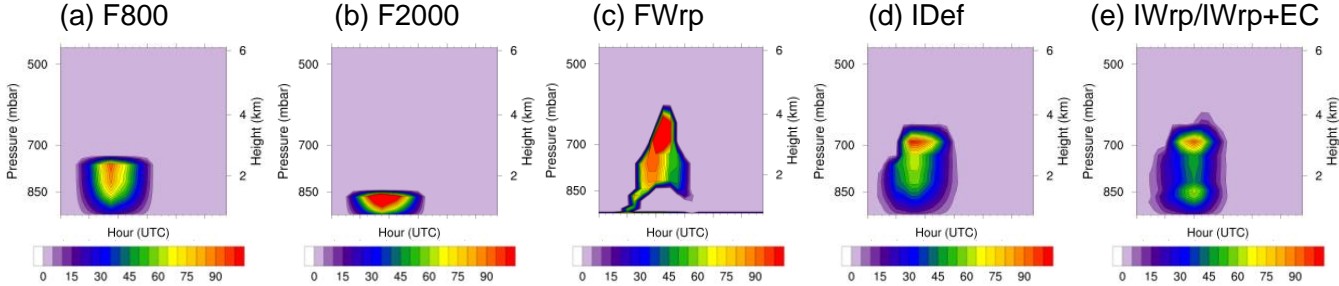


**Figure 2: Initial CO concentration (ppm) profile at Mae Hong Son, Thailand on 13 Mar 2013 (UTC) for each case setup in Table 2 with (a) F800, (b) F2000, (c) FWrp, (d) IDef, (e) IWrp/IWrp+EC.**

## 3 PLMRIM performance

### 3.1 Ground-based measurement stations

The model output is compared with the measurement data at a high-altitude background mountain station in western North Pacific, LABS (receptor; 2,862m AMSL, 23.47°N, 120.87°E) and Doi Ang Khang Meteorology Station (DAK) (source; 1,536m AMSL, 19.93°N, 99.05°E) marked in Fig. 1b,c. The DAK station is an upwind near-source BB location in nPSEA, located in the Chiang Mai Provinces, Thailand, close to the border of Myanmar and Thailand. It is located away from the



cities and mainly received airmass from burning region on the upwind area (Hsiao et al., 2016; Pani et al., 2016) which made
this site representative of the BB emissions from Myanmar, on the western side of Thailand (Khamkaew et al., 2016; Wang
et al., 2015a). The hourly $PM_{2.5}$ data from DAK station is collected during the 2013 7-SEAS spring campaign. Table 3 shows
the performance of PLMRIM on daily $PM_{10}$, daily $PM_{2.5}$, hourly $O_3$ and hourly CO at LABS and DAK according to the
model benchmark (correlation coefficient, R; Mean Fractional Bias, MFB; Mean Fractional Error, MFE) suggested by the
Taiwan EPA (Appendix C). MFB results show that the pollutants are generally over-estimated at these mountain stations.
Unlike the case in the maritime continent that worked best with the F800 method (Wang et al., 2013a), both the fixed height
methods (**F2000**, **F800**) do not apply well for the nPSEA region. Only slight improvement is observed for the offline module
(**FWrp**) with injection height varies according to the fire size. The inline modules (**IDef**, **IWrp**) have obvious improvement
at both LABS and DAK. For the ground stations in Taiwan and Thailand (black markers in Fig. 1b,c), all models have
underestimated the pollutant concentrations while the **IWrp** has performed better than the default inline mechanism with
higher correlation attained. The daily $PM_{10}$ at the North Thailand PCD sources stations for IWrp achieved R=0.84, improved
from R=0.77 of FWrp while daily $PM_{2.5}$ at the Taiwan EPA ground stations for IWrp achieved R = 0.46, improved from
R=0.26 of FWrp (see Table C1 for detail comparison). Adjustment of anthropogenic emission with ECLIPSE data
(**IWrp+EC**) shows clear improvement of CO especially in the stations in Taiwan but not in Thailand. The comparably
insignificant emission amount of anthropogenic emission compared to the BB emission at the near-source BB sites in
Thailand is attributed to the minor pollutant changes during the BB period.

Among all, the inline modules (**IDef**, **IWrp**, **IWrp+EC**) give the lowest bias and closest correlation with the measured
ground station. This highlights the importance of atmospheric stability-based PLMRIM to capture the plume rise variation at
the source site. The boundary layer evolution throughout the day is very much distinctive for mountain-valley compared to
the flat surface where burning usually happens. As highlighted previously (Chuang et al., 2016a; Dong and Fu, 2015b), the
geographical lifting mechanism at the nPSEA is the main factor the BB emission can be carried into the subtropical
westerlies, and hence captured by LABS. Due to the similar performance among the offline and inline settings, the best
performing setup of the offline module (**FWrp**) and inline module (**IWRF+EC**) are selected to simplify the subsequent
discussion.


Figure 3 shows the time series plots for the hourly wind field and $PM_{2.5}$ at DAK source site and hourly wind field, $PM_{10,}$ CO,
and $O_3$ at LABS. The high pollution episode (marked in grey shades) fits well with the great contrast between the model fire
and nofire scenarios and thus confirming that BB plumes are the main pollution source to the high pollution episodes. From
the time series plot, the hourly $PM_{2.5}$ at DAK (Fig. 3a) and hourly $PM_{10}$ (Fig. 3b) at LABS are well captured by the inline
module compared to the offline counterparts. In Fig. 3b, the wind direction shifted to strong south-westerlies in the 2nd half
of March. It is followed by a rise in pollution level at LABS. The offline module (**FWrp**) has significantly overpredicted
$PM_{10}$ at some peaks, even up 200 µg m$^{-3}$. Fair agreement is obtained for CO (Fig. 3c) and $O_3$ (Fig. 3d) with slight





overestimation when concurrent high $PM_{10}$ is modelled. Short-term peak values of 4-5 hours are observed in all models for $PM_{10}$, CO, and $O_3$. The systematic peaks for these pollutants are believed to be the uncertainties involving the FINN BB

emission (Pimonsree et al., 2018). It is found that the performance of $O_3$ is relatively unaffected by the PLMRIM choice.

**Table 3: Performance of modelled chemistry field with different settings of PLMRIM at mountain site in western North Pacific (LABS) and nPSEA (DAK). R: correlation coefficient; MFB: Mean Fractional Bias; MFE: Mean Fractional Error.**

| Parameters | Index | Standard | F2000 | F800 | FWrp | IDef | IWrp | IWrp+EC |
|---|---|---|---|---|---|---|---|---|
| **LABS - Taiwan** | | | | | | | | |
| **Daily $PM_{10}$** | R | x > 0.5 | **0.69** | **0.69** | **0.65** | **0.69** | **0.69** | **0.68** |
| | MFB | -0.35< x< 0.35 | 0.82 | 0.80 | 1.07 | **0.11** | **0.07** | **0.03** |
| | MFE | x< 0.55 | 0.82 | 0.80 | 1.07 | **0.33** | **0.32** | **0.25** |
| **Hourly $O_3$** | R | x > 0.45 | **0.46** | **0.46** | **0.52** | **0.49** | 0.39 | 0.27 |
| **(>40 ppb)** | MNB | -0.15< x< 0.15 | **0.11** | **0.10** | 0.22 | 0.18 | **0.12** | **0.08** |
| | MNE | x< 0.35 | **0.20** | **0.20** | **0.26** | **0.24** | **0.20** | **0.17** |
| **Hourly CO** | R | x > 0.35 | **0.60** | **0.59** | **0.61** | **0.62** | **0.62** | 0.53 |
| | MNB | -0.5< x< 0.5 | **0.51** | **0.50** | 0.63 | **0.45** | **0.43** | **0.29** |
| | MNE | x< 0.5 | 0.55 | 0.55 | 0.66 | **0.50** | **0.49** | **0.38** |
| **DAK- Thailand** | | | | | | | | |
| **Daily $PM_{2.5}$** | R | x > 0.5 | **0.85** | **0.86** | **0.76** | **0.78** | **0.79** | **0.79** |
| | MFB | -0.35< x< 0.35 | 0.59 | 0.59 | 0.53 | **0.29** | **0.35** | 0.36 |
| | MFE | x< 0.55 | 0.63 | 0.62 | 0.61 | **0.32** | **0.38** | **0.38** |




**Figure 3: Comparison of PLMRIM (observation (black), nofire (blue), FWrp (green), IDef (orange), IWrp+EC (red) of (a) hourly wind field and PM$_{2.5}$ at DAK, and (b,c,d) hourly wind field and (b) PM$_{10}$ (b), (c) CO, (d) O$_3$ at LABS in Mar 2013; Grey shade highlights the high pollution hour at LABS (CO > 300 ppb, PM$_{10}$ > 35 μg m$^{-3}$). Wind field for observation (black) and simulation (red) are shown in vector form.**


### 3.2 Aerosol vertical distribution

As illustrated in the shaded region in Fig. 3, the major period that affected LABS is during18–28 Mar 2013. The transport time is known to be around 2–3 days (Chuang et al., 2015), but a longer time of 4–5 days is taken to account for the BB emission generation, lifting, and dispersion at the source site. Hence, the vertical profile of the extinction coefficient from the ground lidar data on MPLNET v0 L1.5a and model output at DAK station during 13–28 Mar is compared in Fig. 4. In Fig.4a, the MPLNET extinction coefficient has shown that the aerosol layer presents around 3–4 km height and mainly confined below the boundary layer. Despite the lower concentration on the surface than the boundary layer, the concentrated amount of aerosol detected by MPLNET (14–15, 19–22, and 23 Mar) still agrees well with the DAK time-series data in Fig. 3a.

The 3-hourly average profile of the extinction coefficient from MPLNET (1 Mar–15 Apr 2014) and **IWrp+EC** and **FWrp** model output (13 – 28 Mar 2013) is illustrated in Fig. 4b-d. The incomplete MPLNET dataset of 2013 has prompted the use of the data from 2014 (Version 2 and Level 1.5) (Wang et al., 2015a) with a similar number of burning hotspots (sum of hotspot covered in model domain 2: 2013 = 1.1 x $10^5$, 2014 = 1.2 x $10^5$) and AOD (averaged from MERRA-2 AOD product in model domain 2: 2013 = 0.34, 2014 = 0.38) during the period of study. In Fig.4b, the MPLNET extinction coefficient increases with height up to around 2.3 km (775 hPa) and reduces around 3.2 km (650 hPa) before hitting another peak around 3.5–4 km. The 2-layer structure is also observed over nPSEA based on multiyear datasets from AERONET study (Feb – Apr, 2007 to 2011(Gautam et al., 2013) and MISR (Feb-Apr, 2001-2010; Jian and Fu, 2014). The model output in Fig. 4c,d shows that the maximum layers above the presumed cap (3.2 km) occurred most prominently during the evening to midnight, and more often in offline than inline modules. The model shows that the offline module gives a time-invariant large value over the entire layers while the inline module is giving a greater approximation on the diurnal variation with the MPLNET result throughout the day. Therefore, during the daytime, the offline module has produced a higher plume height than the CMAQ inline module (Guevara et al., 2014). However, the extinction coefficient of inline output is around 3–4 hours of time lag behind that observed by the MPLNET system.

The extinction coefficient from MPLNET and model output data are only available for qualitative comparison due to their generically different derivations. The lidar system determines the extinction coefficient through the backscatter feedback from the release of the laser beam at 527 nm at every minute, while, the CMAQ model used the mass reconstruction method to sum up the extinction coefficient of each model aerosol species in each layer (Mebust et al., 2003). The empirical assumption for each species and the lower vertical model resolution is attributed to the uncertainties of the modelled extinction coefficient that is typically higher than the value retrieved by MPLNET.

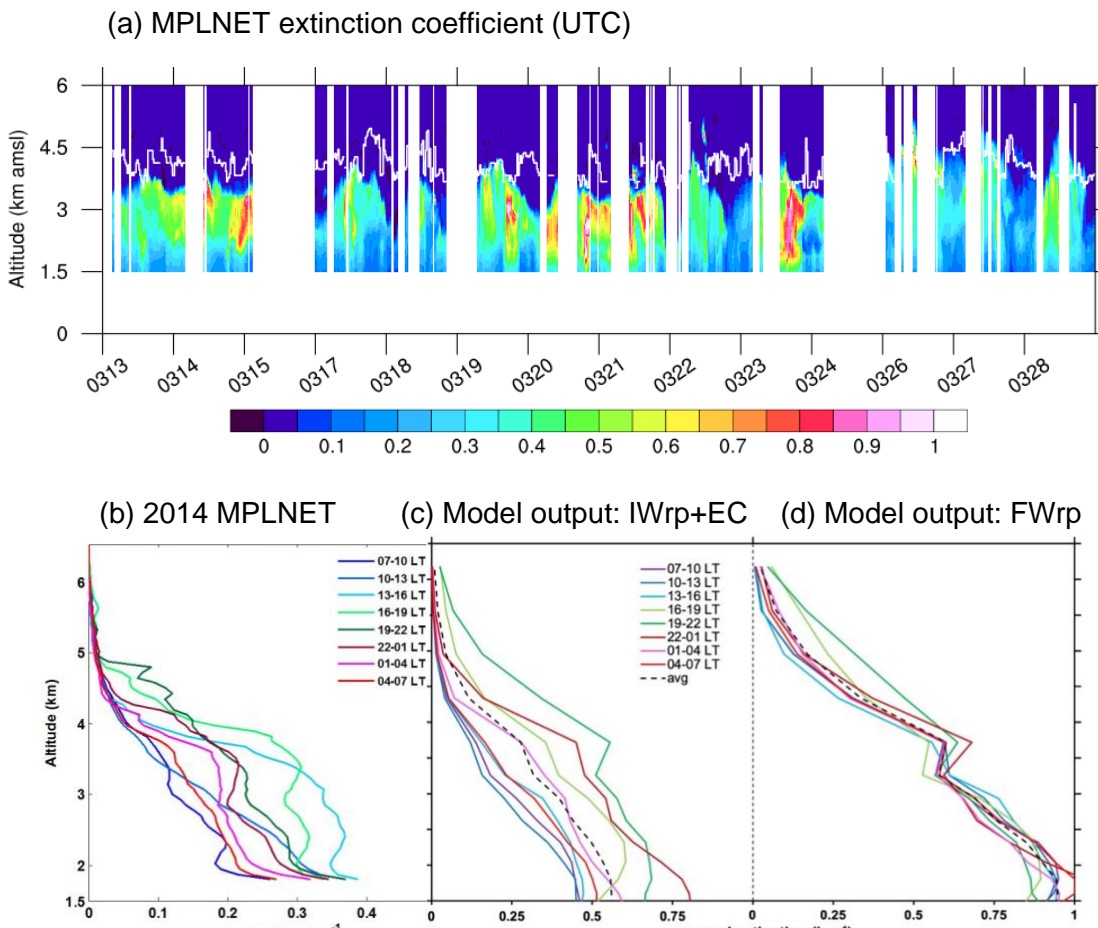

**Figure 4: Vertical extinction coefficient profiles between 13 to 28 Mar 2013 at DAK station from (a) MPLNET with boundary layer height (white); (b) MPLNET 3-hourly average extinction coefficient from 1 Mar – 15 Apr 2014 (Figure 6b directly extracted from** (Wang et al., 2015a))**; Modelled 3-hourly averaged output from 13 Mar – 28 Mar 2013 for (c) IWrp+EC, (d) FWrp.**

Figure 5 shows the CALIOP VFM at the midpoint of BB emission transport route to the receptor during one of the episodes on 19 – 20 Mar 2013. On 19 Mar morning when the sensor (swath: Fig. 5a) captured the smog layer at the height of 4 km above mean sea level (amsl) over the mountainous region (Fig. 5b,c). The aerosols detected are mainly made up of smoke and mixed polluted continental aerosols, which is the main burning emission source. It is known that the burning aerosols from the west part of nPSEA are orographically lifted by west-to-south-westerlies to a higher altitude depending on the terrain height (Cheng et al., 2013; Wang et al., 2015b). For the swath in Fig. 5d – f, the aerosol layers are detected on high levels up to 4 km during the midday. It is most certain to be transported over from the nPSEA since the aerosol layer is detected over the sea where burning does not occur. Secondly, the plume thickness is around 4 km despite the flat land surface, which is much higher than the source site which usually ranges between 0 – 3 km. The aerosol layers are believed to be lifted to a higher level and also mixed to the surface over the land mask in southeastern China. This region locates one of



the largest cities and main industrial bases in Asia, Pearl River Delta (PRD) which produces a large amount of anthropogenic emission. The potential vertical mixing is very likely to pick up the pollutants from the industrial base into the aerosol plume. Recently, it is proven through brute-force methods that the pollution from clusters arrived at the higher altitude in

Taiwan during the winter season (Chuang et al., 2019). About 12 hours later when the swath (Fig. 5g – i) moves closer to Taiwan, the plumes move towards north of 16 ºN but still maintain at a similar altitude that can be detected by the LABS station at 2.4km amsl (Fig. 1). The plume is also found to continue gain in moisture content along the path.





Figure 5: CALIOP vertical feature type and aerosol subtype on continuous episode starting from (a – c) 19 Mar (06:02 LST), (d – f) 19 Mar (13:42 LST), (g – i) 20 Mar (02:07 LST). The corresponding position of the satellite swath is marked in points of red and grey marked in (a,d,f) and altitude below 0 km in (b,c,e,f,h,i). Feature Type: 0 = invalid, 1 = clear air, 2 = cloud, 3 = aerosol, 4 = strato, 5 = surface, 6 = subsurface, 7 = no signal; Subtype of Feature: ND = no data, 1 = marine, 2 = dust, 3 = polluted continental, 4 = clean continental, 5 = polluted dust, 6 = smoke.





A detailed comparison of vertical distribution for all sensitivity tests is given in Appendix D. but here we continue to discuss **FWrp** and **IWrp+EC** cases. In general, the offline **FWrp** produces a much higher concentration of high $PM_{10}$ aerosol layers compared to the inline **IWrp+EC**. Figure 5 shows the model $PM_{10}$ result for **FWrp** (range: 0-300 µg m$^{-3}$) and **IWrp+EC** (range: 0-120 µg m$^{-3}$) for the corresponding period of CALIPSO swath in Fig. 5. Comparison of Fig. 6a-d shows that the **FWrp** produces higher plumes and **IWrp+EC** produces lower plumes since the former produces the initial plume profile on

19 Mar that is consistently high and less dependent on the atmospheric stability induced by mountain flow (Figure D1). Further from the source site (Fig. 6e,f), both runs predict a much lower aerosol layer around 2 km, compared to the 4 km height captured by the CALIOP sensor. The under-representation of both systems along the transport path above sea might be due to the moisture detrainment and entrainment process that is not accounted for in the current model (Paugam et al., 2016; Sofiev et al., 2012).


With a concentration difference of more than 2 times between **FWrp** (up to 300 µg m$^{-3}$)  and **IWrp+EC** (up to 120 µg m$^{-3}$), a more accurate value is captured at LABS by the **IWrp+EC** as shown in Table 3. Regardless of the PLMRIM used, the top height of the plume is confined by an overhead upper-layer wind system. The system has created a strong shear and suppressed the lifting pertaining to the burning convective heat. This explains the invariant of plume height when different

settings are used.

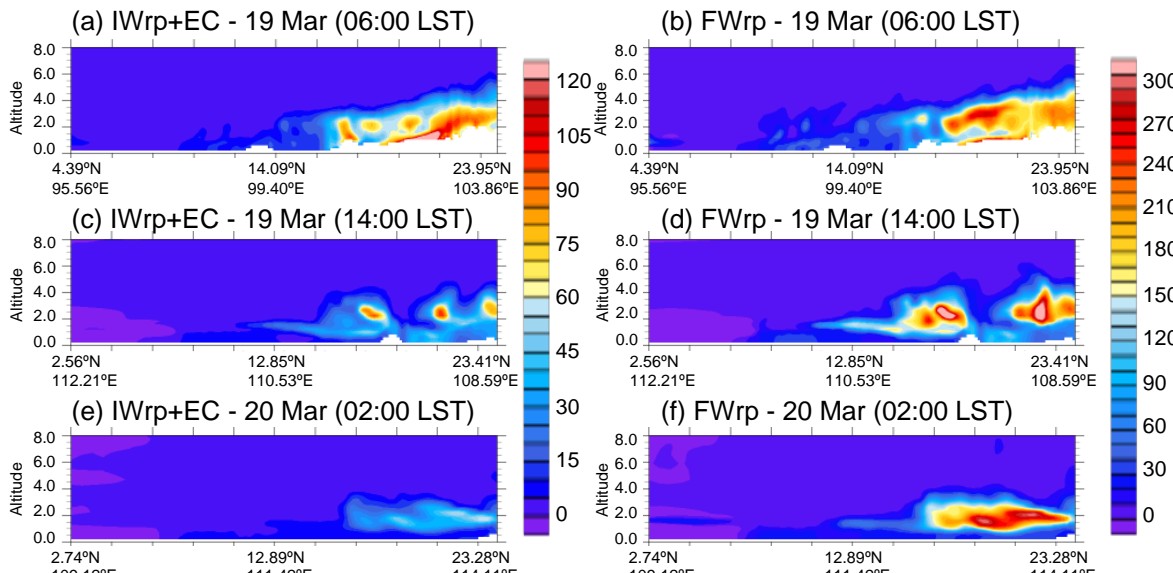

**Figure 6: Comparison of model $PM_{10}$ cross-sectional profile corresponding to CALIPSO period and swath in Figure 4. The range of the left panel is 0–120 µg m$^{-3}$, right panel is 0–300 µg m$^{-3}$.**

The cross-sectional profile in Fig. 6 shows that the amount of emission produced by the offline method is substantially larger

than the amount produced by the inline method. Therefore, the total columnar AOD data provided by 1° x 1° MODIS Terra




Level 3 AOD product (MOD08_D3, Platnick et al, 2015) during the same period (20 Mar 10:30 LST) is used for the verification of the aerosol concentration. Figure 7 shows that the total column AOD produced by the inline module gives a closer approximation to the MODIS. **FWrp** greatly overestimates the aerosol produced by the BB emissions, while the inline module gives a closer agreement on northern Thailand and southern Vietnam.

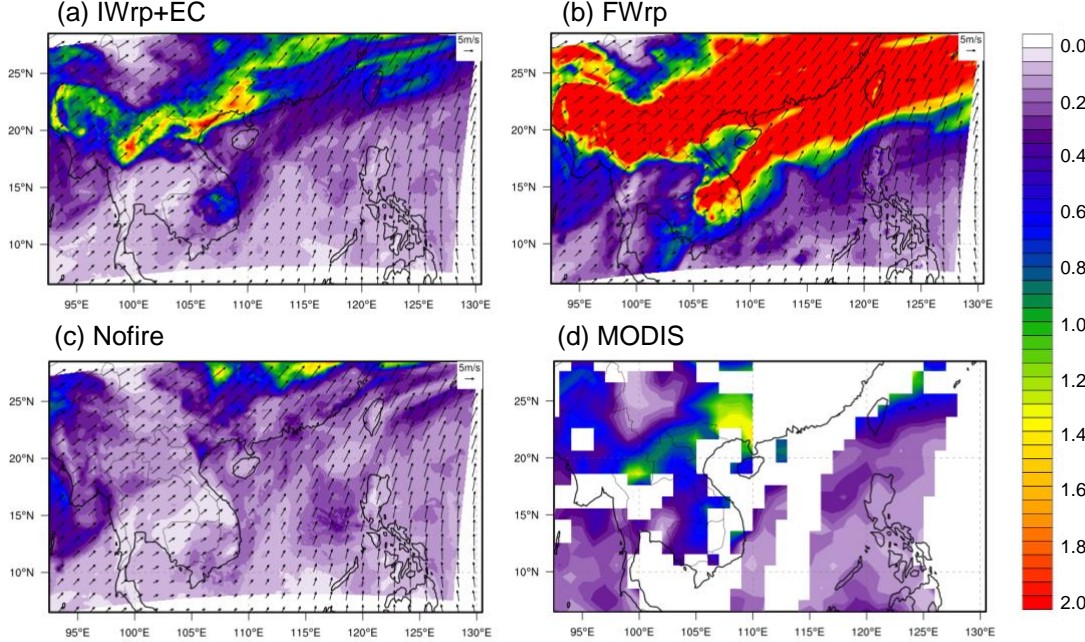


**Figure 7: Comparison of daily total column AOD on 20 Mar (10:30 LST) of model output (a) IWrp+EC, (b) FWrp, (c) Nofire with (d) MODIS data from Figure 5. Vector profiles given in (a-c) are the surface wind profile.**

### 3.3 Reliability of inline PLMRIM

The variation of model performance has intrigued the compatibility of emission inventory with the PLMRIM performance.
The FINN dataset provides high-resolution data for each fire (1 km$^2$) and would be more representative in the inline calculation that is proceeded with the plume-in-grid concept. Therefore, if the offline method is adopted (**FWrp**), the high-resolution emission dataset FINN in the nPSEA region tends to over-predict by 4-fold (Fig. 3a). Previous literature has to make an adjustment to the fire inventory to bring down the FINN emission amount that was overestimated by up to 2-3 times of PM$_{2.5}$ and PM$_{10}$ at the source region (Pimonsree et al., 2018), and FLAMBE overestimates up to 3 times for CO and PM$_{10}$
at the LABS site (Chuang et al., 2015; Fu et al., 2012). In this paper, the model discovers that the direct application of the FINN dataset is able to work well with the inline module (**IWrp+EC**). BB emission is mainly caused by small fires and dry conditions over the period in the region (Giglio et al., 2013; Reid et al., 2013), this also explains why the inline module worked well to represent the BB condition.





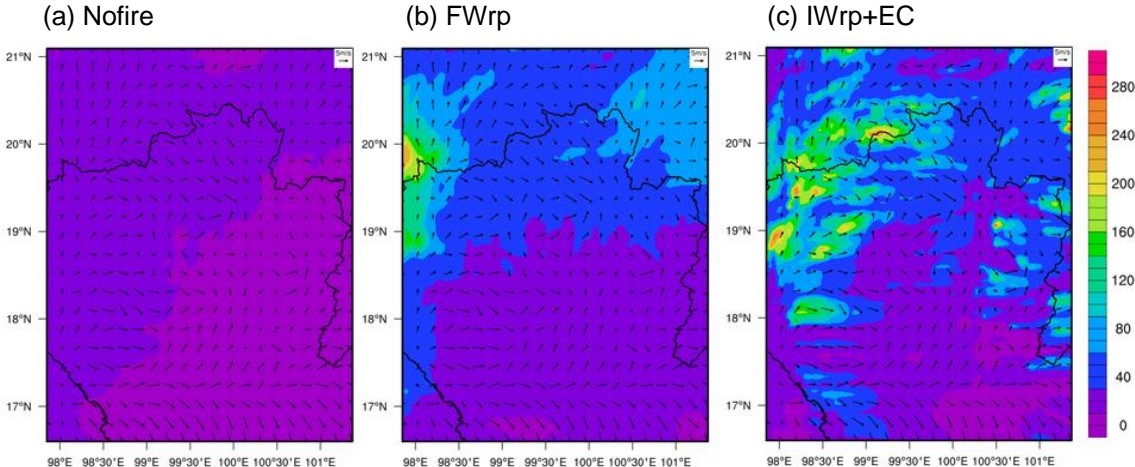

**Figure 8: Spatial distribution of PM$_{10}$ concentration on 19 Mar 17:00 LST over burning regions of nPSEA for 4$^{th}$ domain (d04)**

The inaccuracy of the offline module is likely to be caused by the role of the complex terrain in uplifting the smoke plume and the nature of the fuel loadings. The connecting slopes (0.2–1.8 km as seen in Fig. 1c) causes the complication to boundary layer physics that governs the dynamics to transport the plumes formed in the valley pockets. Due to the unique topographic structure in nPSEA, the lifting and breaking away of burning emission plumes from burning area occurs during the evening-to-night period. Therefore, mountain meteorology played an important role in the distribution of higher-level plumes. Moreover, the ability of PLMRIM to capture the boundary layer physics becomes essential in the mountainous region. Through the inline module with the WRAP initial plume profile (**IWrp+EC**), the natural buoyancy of fire together with the convective interaction of the atmosphere can correctly distribute the BB emission. The spatial distribution of PM$_{10}$ over burning regions in nPSEA is shown, with comparison made for scenarios nofire (Fig. 8a), offline (Fig. 8b) and inline (Fig. 8c). Comparison of the figures shows that each sub-grid scale fire hotspots more realistically represents the actual high concentration of emission emitted at the source (Fig. 8c) compared to the grid-following averaged out effect in the offline method (Fig. 8b). Nevertheless, the current setting does not include the two-way aerosol-radiation and aerosol-radiation-cloud feedback. This will be further studied in the future work looking at its importance in the cloud-laden SEA region (Tsay et al., 2016), as seen in the missing data due to the cloud cover in Fig. 6d.

## 4 Transport of biomass burning aerosol to Taiwan

The below discussion is performed using the model output of **IWrp+EC** and focuses on the high pollution episodes observed at LABS during 13–28 Mar 2013 as seen in the grey shaded area of Figure 3. In the source region of nPSEA, the complex land terrain has played a substantial role in the BB plume lifting. Figure 9 shows the evolution of the PM$_{10}$ concentration on 13 Mar 2013 at DAK but over the nPSEA through the cross-sectional profile (Fig. 1c). During the day





when the fires are active, BB emission is released from the surface (Fig. 9a, b). Along with the rising of planetary boundary layer height (PBLH), the BB aerosol mixes into the entire boundary layer. The residue layer starts to form during the transitional period between the day and night around 17:00 LST (Fig. 9c) when the ground surface cools down. When the

atmosphere becomes stable into the night, the aerosol layer remains as the residue layer and does not move down with the boundary layer (Fig. 9d). The plume starts to be advected by the shear of the upper layer flow at night on the downwind leeside of the hills. It is because the boundary layer height tends to rise higher due to turbulence. The descent of the boundary layer also confines the aerosol and causes a high concentration near the surface. The detachment of the aerosol layer therefore explains the two-layer plume feature from evening into the night in Fig. 4b,c. The dispersion of emission

from the pockets is subjected to at least three systems, (i) strong westerlies from Myanmar flowing over the top of valley pockets that confined the emission (terrain structure shown in grey in Fig. 9), (ii) diurnal mountain-valley breeze might trap or disperse the emission, (iii) local heating caused by the solar cycle affects the plume rise and disperse the emission. Therefore, the amount of burning emission lifted is greatly coherent with the populated hills along the transport path.

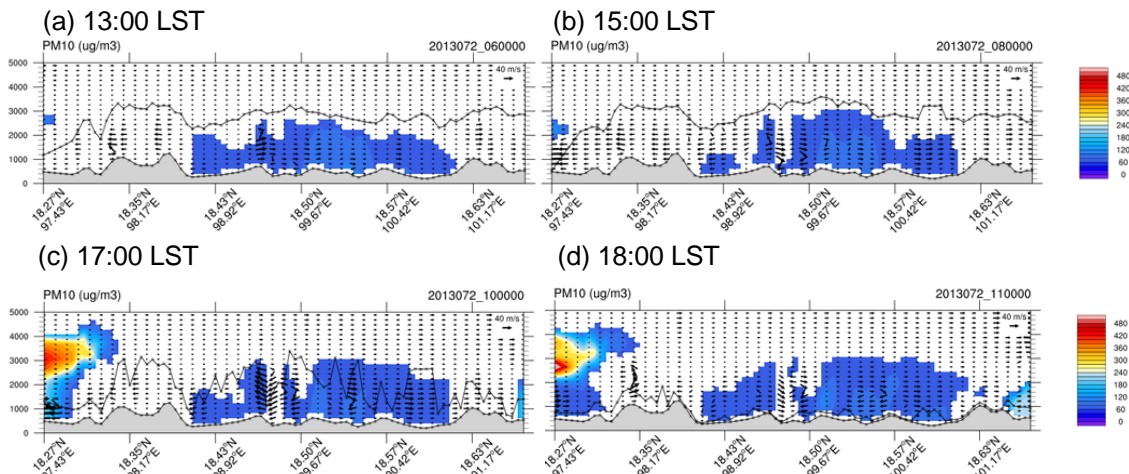

**Figure 9: The modelled vertical cross section profile (BB' in Figure 1c) up to 5 km over nPSEA on 13 Mar: (contour) PM$_{10}$ concentration (IWrp+EC, µg m$^{-3}$), (vector) horizontal wind profile (ms$^{-1}$) in x-direction and vertical wind profile in y-direction (cm s$^{-1}$), (dotted lines) boundary layer height in meter, (shaded) terrain.**

Comparing the model output data of the inline (**IWrp+EC**) and nofire, Figure 3 shows that BB from nPSEA contributes 68±18% to PM$_{10}$, 66±18% to PM$_{2.5}$, 41±13% to O$_3$ and 58±13% to CO during the intense BB period (18 – 27 Mar) to LABS.

While BB contributes 43±31% to PM$_{10}$, 41±32% to PM$_{2.5}$, 23±19% to O$_3$, and 39.1±23.0% to CO at LABS for the entire month of Mar 2013. The transport pathway of BB from nPSEA to LABS coincides with the anthropogenic emissions from the nPSEA as well as the southeast China, BB aerosols from such emission region are also captured in the model. Therefore, the actual amount might indicate a slightly lower contribution by BB aerosol than the derived contribution. There are several mechanisms identified in Mar 2013 to bring BB smoke to Taiwan.




**4.1 Westerlies to carry BB emission to LABS**

In this case, the BB aerosol lifted is further carried by strong westerlies on the upper layer, around height between 2–4 km towards LABS. This usually occurred during the night when the atmospheric boundary layer is low and stable as shown in Fig. 10. This is the commonly known mechanism that carries the BB plumes to higher ground in Taiwan. This condition occurred on 19–20, 24–25, 27–28 Mar 2013. This is the commonly known scenario that is well studied due to the availability

of measurement collected at LABS.

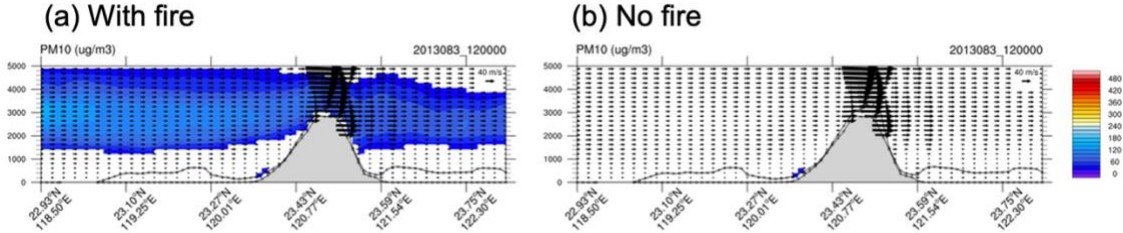

**Figure 10: Vertical cross-sectional AA' (Figure 1b) profile for PM₁₀ (contour), wind at x-z direction (vector), PBLH (dotted lines) and terrain height (grey shade) on 2013083 12:00 UTC (24 Mar 20:00 LST) for (a) with fire, (b) no fire.**

**4.2 Mixing of BB emission with local pollution on surface**

The land surface is heated up and the boundary layer during the day grows as high as 1.5 – 2 km on western Taiwan, around 1 km on the windward of the central mountain range, and up to 4 km amsl at LABS. When the BB plumes overpass are as low as the BLH, then the BB aerosol is brought into the boundary layer and mixed to the ground as shown in Fig. 11. The interaction of BB with local pollutants depends on the loading of local pollutants present. The latter is subjected to the local weather system and the occasional Asian continental cold surge that might clean the accumulated pollutants. Such cases

usually occur during the morning to noontime when the land surface heats up and PBLH develops. This condition occurred on 18, 19, 20, 21, 28 Mar 2013. This is the main mechanism that affects the western Taiwan. It was pointed out that cold surge might be responsible for the downdraft of the BB smoke plumes to the surface (e.g. Lin et al., 2017).



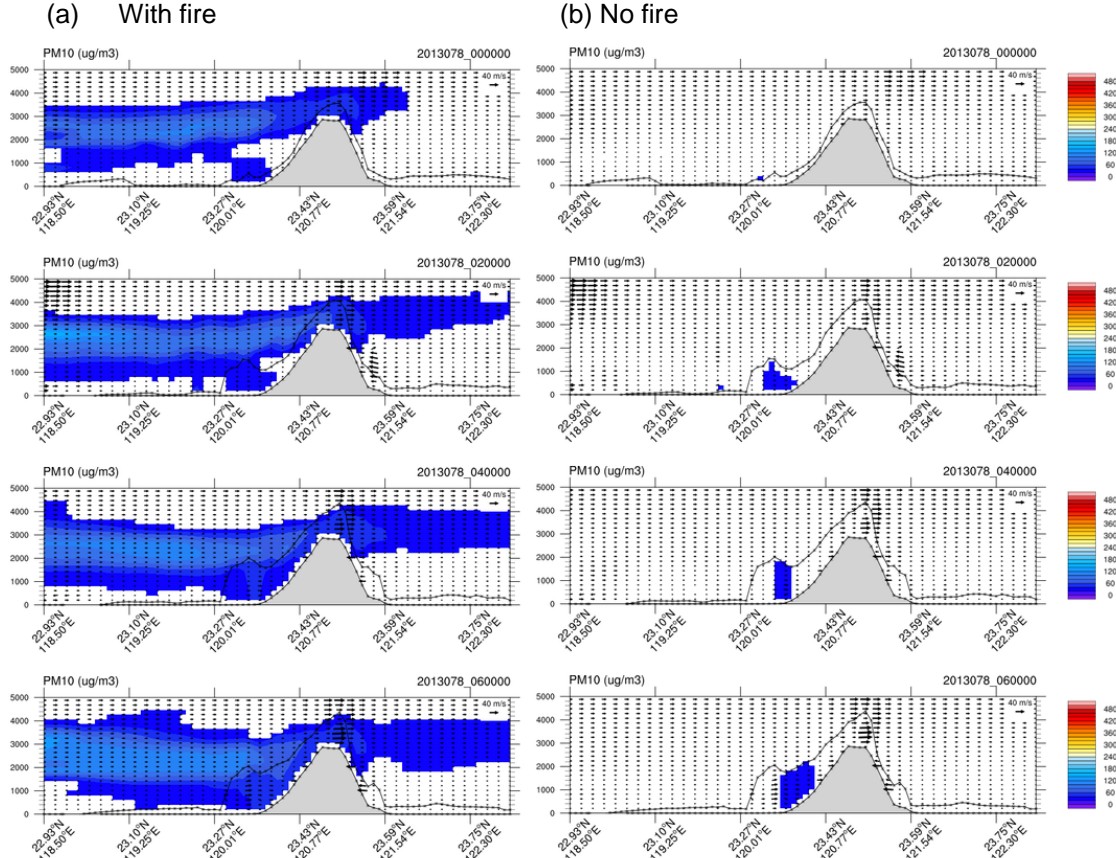

Figure 11: Similar to Figure 10 but on 2013078 00:00, 02:00, 04:00, 06:00 UTC (24 Mar 08:00, 10:00, 12:00, 14:00 LST) for (a) with fire, (b) no fire.

## 4.3 Mixing of BB emission with local pollution above surface

Along with the sea-land heat difference, the sea breeze and mountain breeze are formed and enhance the uphill movement of local pollution in western Taiwan. In such a case, the local pollution is brought up to a high elevation to interact with the BB smoke plumes as shown in Fig. 12. It also occurred that the local pollutants brought uphill detaches from the planetary boundary layer when the surface cools down quickly. This residue layer of pollutants is then mixed into the BB layers and carried towards the east. Such cases usually occur during midday when the local pollution plumes have moved up to the hill. This condition occurred on 17, 23, 25 Mar. The detection of BB intrusion into surface sites in southwestern Taiwan is not a rare occasion (Huang et al., 2013; Tsai et al., 2012). A larger amount of fine nanoparticles from local sources is measured at LABS especially during the morning even not during the spring season (Chen et al., 2013). Therefore, it is possible that mixing does occur when the local pollutants are transported up the hill through the valley breeze.

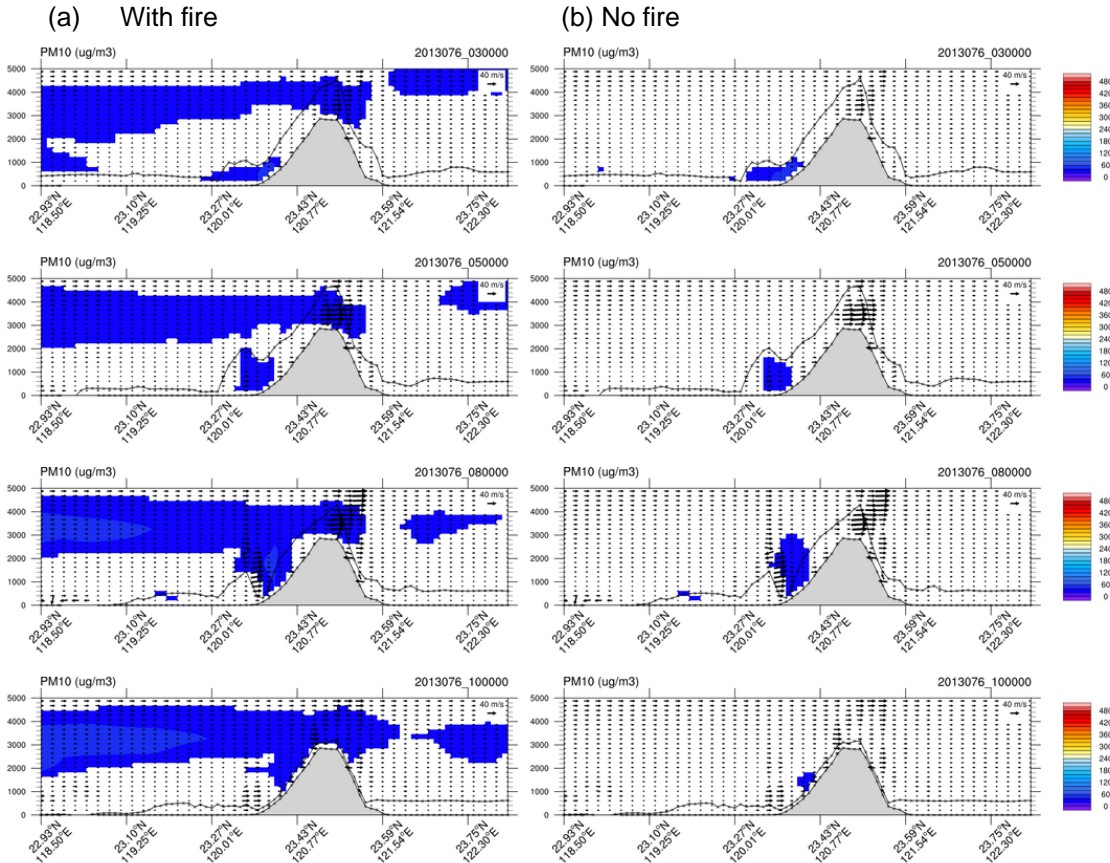

**Figure 12: Similar to Figure 10 but on 2013076 03:00, 05:00, 08:00, 10:00 UTC (22 Mar 11:00, 13:00, 16:00, 18:00 LST) for (a) with fire, (b) no fire.**

Among the three mechanisms, the BB aerosols have the most direct influence on the surface site in western Taiwan, which is coherent to the reduction of surface $O_3$, $NO_x$, and $SO_4^{2-}$ aerosols in 2006 (Dong et al., 2018). However, all these three mechanisms are prone to alter the radiative forcing over western Taiwan. The future incorporations of the aerosol radiative forcing effect through one-way and two-way meteorology-chemistry process of moisture detrainment and entrainment are necessary to understand the role of BB aerosol on the weather extremes in downwind regions. The cloud-aerosol interaction is particularly crucial to the study of the impact of BB aerosols on cloud-laden regions between nPSEA and Taiwan (Hsu et al., 2003; Tsay et al., 2016). The allocation fraction will need to improve looking at the importance of small fire smoldering in SEA (Akingunola et al., 2018; Zhou et al., 2018).





## 5 Conclusion

In this study, several factors involved in the modelling of BB smoke plumes are tested in the WRF-CMAQ model, namely the injection height, initial vertical distribution profile of BB emission, inline PLMRIM, and amount of anthropogenic emission. The conventional method used for the study region adopted the fixed height allocation which produces an excessive amount of emission over the entire transport route. The initial vertical allocation profile according to the WRAP empirical coefficient (**IWrp**) improves the surface concentration of the BB emission by the inclusion of the smoldering fraction compared to the default inline PLMRIM (**IDef**). While replacing the emission in SEA countries from MIX (**IWrp**) to ECLIPSE (**IWrp+EC**) also improves the pollution concentration simulation at the downwind LABS, especially CO which is the important tracer of anthropogenic emission.

The model comparison shows that regardless of the injection height, the main deficiency of the fixed height offline algorithm originates from its invariant vertical-layer allocation of BB concentration throughout the day. In the complex terrain over the nPSEA region which is continuous and varies between 0.2 km to 1.8 km, mountain meteorology played an important role in the distribution of higher-level plumes. The two-layer structure of the BB plumes observed in the MPLNET extinction coefficient profile at night is well captured by the inline PLMRIM (**IWrp+EC**) while the offline method (**FWrp**) gives a time-invariant large value over the entire layers. This highlights that the inline PLMRIM (**IWrp+EC**) is able to incorporate the diurnal boundary layer physics of the mountain to accurately represent the vertical distribution of the BB concentration in the source and downwind region. It is then clear that the amount of emission produced by the inline reasonably captures the columnar AOD distribution over the transport route between nPSEA and downwind Taiwan when compared to the MODIS columnar product. It is discovered that the inline module with the initial distribution profile of WRAP (**IWrp+EC**) is able to and performs well both at the source and receptor sites compared to the offline module.

The model output shows that the BB plumes near nPSEA are emitted during the day within the BLH. Due to strong mountain-valley wind, the smoke plume layers tend to detach from the BLH as residue layers when the surface cools down in the evening-to-night period. This is the layer of plumes that entered the free troposphere at approximately 1-3km height and further transported over to western north Pacific and Taiwan. The plume layers clearly affect the Taiwan region via three conditions: (a) overpass western Taiwan and enter mountain area (LABS), (b) mix down to western Taiwan, (c) transport of local pollutants up and mix with BB plume on LABS. The second condition involves the prevailing high-pressure system that is able to impact the most population in Taiwan and would be an interesting case to explore.

However, care should be taken to select the BB emission inventory input when switching from the offline module to the inline module. The sub-grid scale allocation of the BB emission requires higher resolution of BB emission inventory such as FINN to reproduce the individual fires with distinct and realistic peaks. The work highlights the importance of atmospheric



stability-based PLMRIM and the accurate application of emission inventories to capture the plume rise variation at the source site with complex terrain. The correct representation at the nPSEA source site substantially affects the downwind BB

concentration in mountain (LABS) and surface sites in Taiwan. It is also observed that the improved setting is able to represent the source site's vertical profile well, however, the height of the plume is reduced following the transport and evolution of the plume approaching Taiwan. This might be caused by the missing algorithm of the indirect and direct effect between aerosols and the high cloud cover region along the transport path. It leads to future exploration and incorporation of the effect of cloud-aerosol interaction over the cloud-laden region.






**Appendix A. Model verification for modelled weather field**

In the following formulas, $M_i$, $O_i$, $\bar{O}$ represent simulated value of record $i$, observed value of record $i$, mean of observed values for $I$ to $N$. $N$ are total number of records.

$$\text{Mean Bias (MB): MB} = \frac{1}{N}\sum_{i=1}^{N}(M_i - O_i)$$


$$\text{Mean Absolute Error (MAE): MAE} = \frac{1}{N}\sum_{i=1}^{N}|M_i - O_i|$$

$$\text{Root Mean Square Error (RMSE): RMSE} = \left[\frac{1}{N}\sum_{i=1}^{N}(M_i - O_i)^2\right]^{\frac{1}{2}}$$

Wind Normalized Mean Bias (WNMB) : $\text{WNMB} = \frac{1}{N \times 360°}\sum_{i=1}^{N}(M_i - O_i) \times 100\%$

Wind Normalized Mean Error (WNME): $\text{WNME} = \frac{1}{N \times 360°}\sum_{i=1}^{N}|M_i - O_i| \times 100\%$

The boundary condition data in WRF model uses the reanalysis weather data. These data are assimilated with measurement
data, they are available in coarse resolution (1° x 1°). The work has hence included the observation nudging settings to
improve its prediction of local area. The data used for nudging are given in Section 2. The assimilation with the default
setting does not improve the prediction hourly T2 and WS, hence the subsequent effort is to adjust the area of influence of
each the measuring stations. The radii of influence (RIN) for both d03 and d04 are updated to 100 km based on the average
distance between the observation stations (d03: 125 km, d04: 153 km) and minimum distance between 2 stations (d03: 64
km, d04: 36 km). Although the wind direction is greatly improved with the modification of RIN, the positive bias of T2 and
negative bias of WS is still apparent, especially for the LABS station. Given that the 3rd domain is of 5 km x 5 km resolution,
the height of Mt. Lulin might be averaged out by the lower terrain surrounding it and the model height of Mt. Lulin is lower
(2216 m, layer = 1) than its original height (2862 m). Comparison has found that model layer 4 from surface is most
representative of the height of Mt Lulin (2492 m; 757 hPa). Hence with the extraction of new location of Mt Lulin, the
prediction of T2 and WS are improved significantly as tabulated in $Table_VERmet. The wind profile over LABS, one of
the decisive weather factors of transport, has complied well with the observation data as seen in Figure 2. The passing rate of
surface cwb stations for hourly T2, WS and WD are also well above the model benchmark (60%).

Table A1: The performance of each stations for weather parameters (T2, WS, WD) in March 2013 for Thailand (TH)
stations, Taiwan (TW) stations, and Lulin (LABS). *Distance given is the radius of influence in observation nudging.
#Station output is extracted from the corresponding model layer of the station height in the model.

| Parameter | Index | Standard | no fdda | fdda; 240 km* | fdda; 100 km*# |
|---|---|---|---|---|---|
| **TH stations** | | | | | |
| **T2** | MB | -1.5< x< 1.5 | -0.3 | -0.3 | -0.3 |
| | MAE | x< 3 | 2.2 | 2.2 | 2.2 |
| **WS** | MB | -1.5< x< 1.5 | 1.2 | 1.2 | 1.2 |
| | RMSE | x< 3 | 1.7 | 1.8 | 1.8 |
| **WD** | WNMB | -10< x < 10 | 2.1 | -4.0 | -4.1 |
| | WNME | x< 30 | 29.5 | 23.4 | 23.3 |
| **TW stations** | | | | | |
| **T2** | MB | -1.5< x< 1.5 | 0.5 | 0.2 | 0.2 |



| | MAE | x< 3 | 2.1 | 2.0 | 2.0 |
|---|---|---|---|---|---|
| **WS** | MB | -1.5< x< 1.5 | 0.5 | 0.7 | 0.7 |
| | RMSE | x< 3 | 1.9 | 1.9 | 1.9 |
| **WD** | WNMB | -10< x < 10 | -4.5 | -9.9 | -10.2 |
| | WNME | x< 30 | 26.6 | 20.8 | 20.9 |
| **LABS** | | | | | |
| **T2** | MB | -1.5< x< 1.5 | 1.6 | 2.3 | **0.2** |
| | MAE | x< 3 | 2.6 | 2.9 | **1.5** |
| **WS** | MB | -1.5< x< 1.5 | -2.6 | -1.9 | **0.9** |
| | RMSE | x< 3 | 3.5 | 3.0 | **2.3** |
| **WD** | WNMB | -10< x < 10 | **0.3** | **-4.0** | **3.4** |
| | WNME | x< 30 | **12.6** | **12.7** | **8.9** |

## Appendix B. Comparison of ECLIPSE and MIX anthropogenic emission

The anthropogenic dataset, ECLIPSE and MIX for year 2010 is compared in Figure B1 for peninsular SEA and in Figure B2 for the entire Asia. Figure B1 shows that ECLIPSE generated lower amount of CO and VOC and higher amount of particulate matters and $NO_x$ over peninsular SEA compared to the MIX dataset. The ECLIPSE data give a higher total $NH_3$,

BC, $PM_{2.5}$, $NO_x$, $PM_{10}$ by 192%, 51%, 38%, 29%, 24% respectively, while lower total VOC, CO, OC, $SO_2$ by 40%, 23%, 22%, 20% respectively. Largest biases are observed in developing SEA countries as seen in Figure B2, such as Laos, Burma, Philippines and Timor-Leste where local data are not easily available. However, the emissions for China and Taiwan are kept unchanged due to the high confidence and quality of respective national emission inventories (Li et al., 2018).

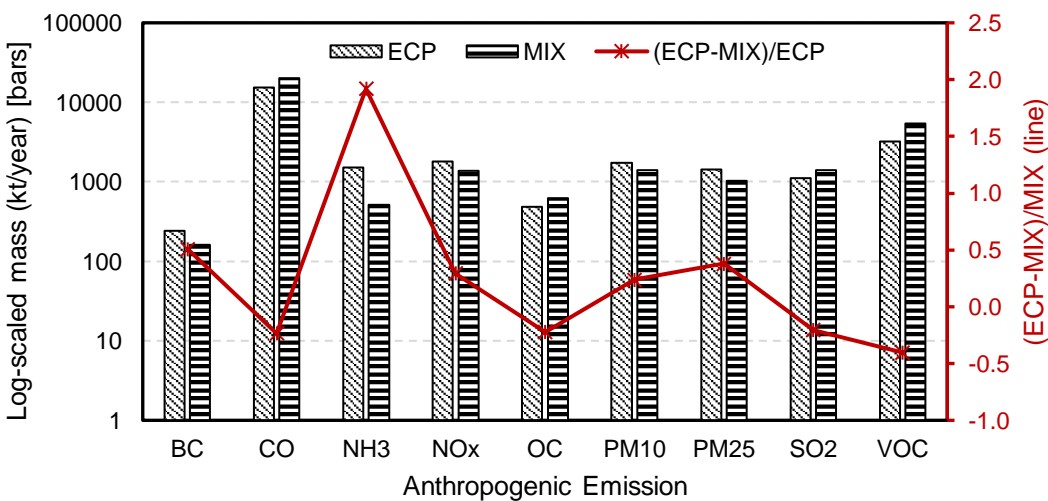

**Figure B1: Comparison of total mass of emitted air pollutants (BC, CO, $NH_3$, $NO_x$, OC, $PM_{10}$, $PM_{2.5}$, $SO_2$, VOC) from anthropogenic emission inventories over peninsular SEA (including Thailand, Vietnam, Cambodia, Burma and Laos) in year 2010: ECLIPSE (ECP; box with diagonal lines), MICS-ASIA (MIX; box with horizontal lines), and difference fraction between ECP and MIX ((ECP-MIX)/MIX); red line).**



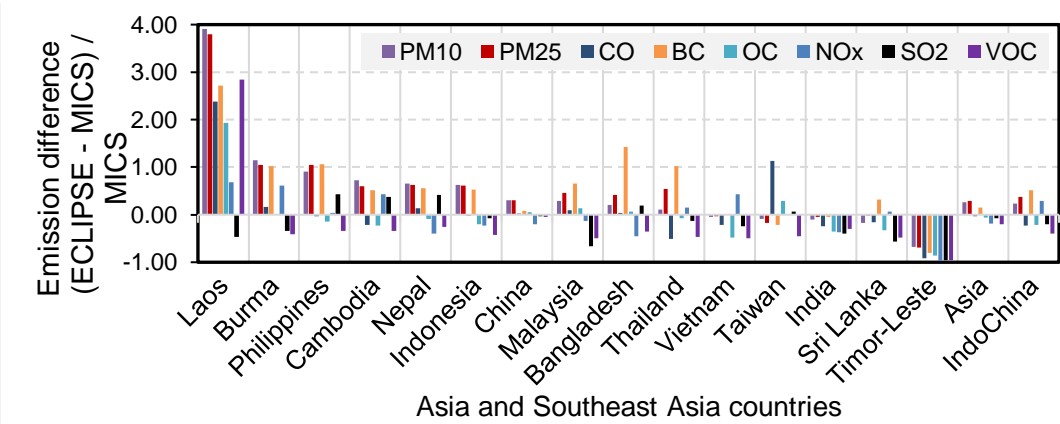

**Figure B2: Comparison of 2010 ECLIPSE and MIX emission in Southeast Asia and Asia countries that are covered within d02, including Taiwan and China.**

### Appendix C. Model verification for modelled air quality

In the following formulas, $M_i$, $O_i$, $\bar{O}$ represent simulated value of record $i$, observed value of record $i$, mean of observed values for $1$ to $N$. $N$ are total number of records.


$$\text{Correlation Coefficient (R): R} = \frac{1}{N-1}\sum_{i=1}^{N}\left[\frac{(M_i-\overline{M})(O_i-\overline{O})}{Stdev_M Stdev_o}\right]$$

$$\text{Mean Fractional Bias (MFB): MFB} = \frac{1}{N}\sum_{i=1}^{N}\frac{M_i-O_i}{(M_i+O_i)/2}$$

$$\text{Mean Fractional Error (MFE) : MFE} = \frac{1}{N}\sum_{i=1}^{N}\frac{|M_i-O_i|}{(M_i+O_i)/2}$$

**Table C1: Performance of modelled chemistry field with different setting of plume rise model at other EPA stations in Taiwan and PCD stations in NT**

| Parameter | Index | Standard | F2000 | F800 | FWrp | IDef | IWrp | IWrp+Ec |
|---|---|---|---|---|---|---|---|---|
| **TW stations (EPA)** | | | | | | | | |
| **Daily PM$_{10}$** | R | x > 0.5 | 0.22 | 0.22 | 0.17 | 0.34 | 0.34 | 0.30 |
| | MFB | -0.35< x< 0.35 | -0.35 | -0.36 | **-0.26** | -0.70 | -0.71 | -0.79 |
| | MFE | x< 0.55 | 0.60 | 0.60 | 0.58 | 0.74 | 0.75 | 0.81 |
| **Daily PM$_{2.5}$** | R | x > 0.5 | 0.30 | 0.30 | 0.26 | **0.48** | **0.49** | **0.46** |
| | MFB | -0.35< x< 0.35 | **-0.11** | **-0.12** | **-0.02** | -0.57 | -0.58 | -0.61 |
| | MFE | x< 0.55 | **0.44** | **0.43** | **0.44** | 0.61 | 0.61 | 0.64 |
| **Hourly O$_3$ (>40 ppb)** | R | x > 0.45 | **0.58** | **0.58** | **0.57** | **0.55** | **0.55** | **0.61** |
| | MNB | -0.15< x< 0.15 | **0.09** | **0.08** | **0.09** | **0.10** | **0.09** | **-0.01** |
| | MNE | x< 0.35 | 0.22 | 0.22 | 0.22 | 0.22 | 0.22 | 0.21 |
| **Hourly CO** | R | x > 0.35 | 0.24 | 0.24 | 0.24 | 0.24 | 0.24 | 0.29 |
| | MNB | -0.5< x< 0.5 | **0.14** | **0.14** | **0.18** | **0.11** | **0.11** | **0.09** |
| | MNE | x< 0.5 | 0.55 | 0.55 | 0.56 | 0.56 | 0.56 | 0.56 |
| **NT Stations (PCD)** | | | | | | | | |
| **Daily PM$_{10}$** | R | x > 0.5 | **0.76** | **0.75** | **0.77** | **0.83** | **0.84** | **0.84** |
| | MFB | -0.35< x< 0.35 | -0.40 | -0.45 | **-0.30** | -0.91 | -0.86 | -0.85 |
| | MFE | x< 0.55 | 0.60 | 0.64 | **0.50** | 0.91 | 0.87 | 0.86 |





| | | | | | | | |
|---|---|---|---|---|---|---|---|
| **Hourly O₃** | R | x > 0.45 | 0.44 | 0.44 | 0.45 | **0.47** | **0.49** | **0.49** |
| **(>40 ppb)** | MNB | -0.15< x< 0.15 | **-0.04** | **-0.07** | **-0.01** | 0.27 | 0.22 | 0.23 |
| | MNE | x< 0.35 | **0.25** | **0.25** | **0.24** | 0.39 | 0.37 | 0.37 |
| **Hourly CO** | R | x > 0.35 | **0.41** | **0.42** | **0.37** | **0.41** | **0.45** | **0.45** |
| | MNB | -0.5< x< 0.5 | -0.50 | -0.51 | **-0.48** | **-0.25** | **-0.21** | **-0.21** |
| | MNE | x< 0.5 | 0.74 | 0.74 | 0.74 | 0.74 | 0.74 | 0.74 |

**Appendix D. Detailed comparison of vertical distribution**

For offline methods, higher plume rise height and concentration vary positively with the initial allocated height (Table 2), with increasing order of F800, F2000 to FWrp. Inline method is generally lower in amount and the near surface emission has increased with IWrp compared to IDef (Figure S2).

(a) F2000                             (b) F800

(c) IDef                                (d) FWrp

(e) IWrp                                (f) IWrp + EC

(g) nofire

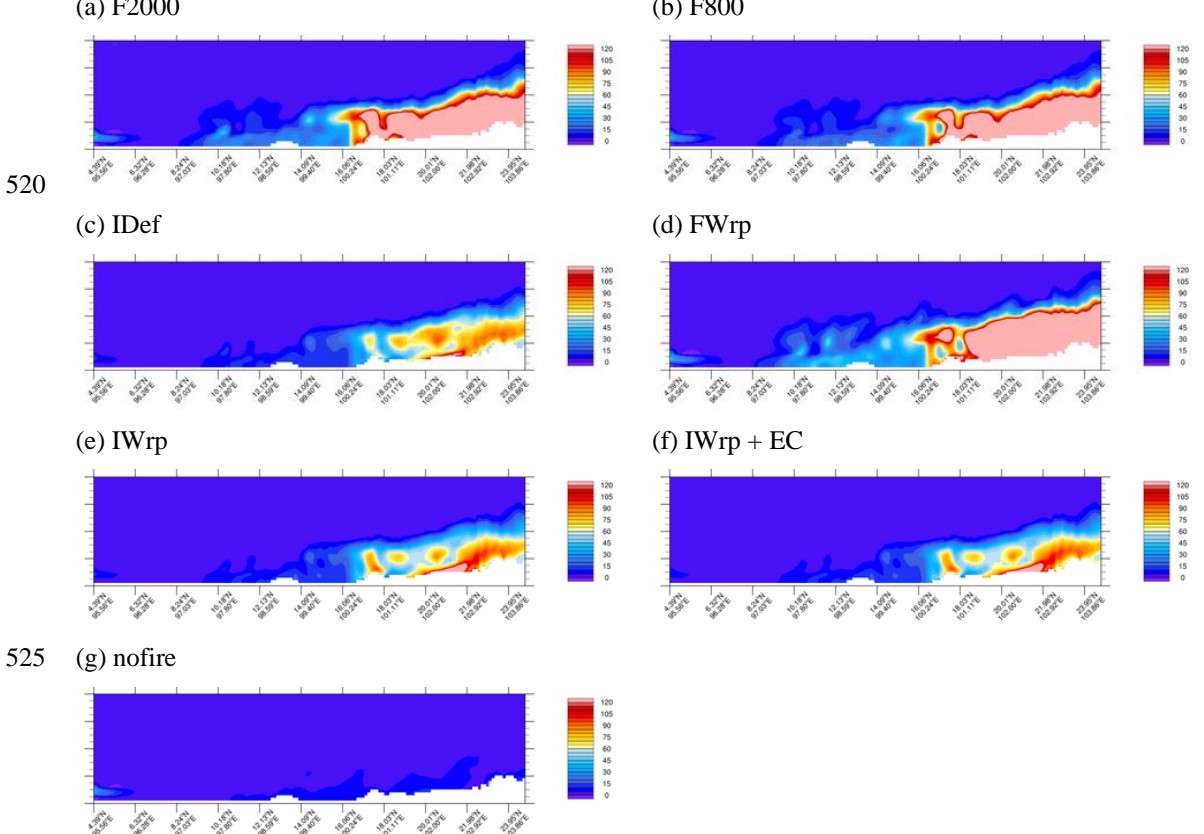

**Figure D1: Comparison of vertical cross-sectional area on 19 Mar (06:00 LST) modelled by each plume rise setting with the same contour scale range (0 – 120 ug.m⁻³)**




(a) IWrp+EC at 20 Mar 17:00 LST    (b) FWrp at 20 Mar 17:00 LST

(c) IWrp+EC at 20 Mar 18:00 LST    (d) FWrp at 20 Mar 18:00 LST

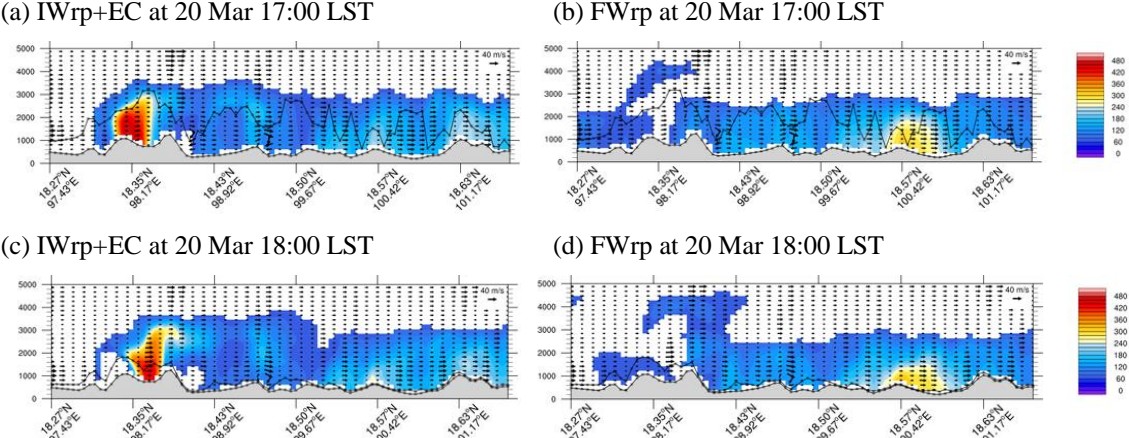

**Figure D2: Vertical PM$_{10}$ BB' cross section (Figure 1c) up to 5 km at NT (d04) on 20 Mar (a) IWrp+EC at 17:00 LST, (b) FWrp at 17:00 LST, (c) IWrp+EC at 18:00 LST, (d) Fwrp at 18:00 LST.**

## Data availability

All the data sets presented in this study are available upon request from the corresponding author.

## Author contribution

Maggie C. Ooi: Data curation, Formal analysis, Investigation, Software, Validation, Visualization, Writing – original draft preparation, Writing – review and editing; Ming-Tung Chuang: Supervision, Investigation, Writing – review and editing Joshua S. F: Conceptualization, Investigation, Supervision, Writing – review and editing; Steven S. Kong: Investigation; Wei-Syun Huang: Project Administration, Data Curation, Software; Sheng-Hsiang Wang: Data Curation, Validation; Andy Chan: Writing – review and editing; Shantanu K.Pani: Writing – review and editing; Neng-Huei Lin: Conceptualization, 545 Investigation, Funding Acquisition, Supervision, Resources, Writing – review and editing





**Competing Interest**

The authors declare that they have no conflict of interest.

**Acknowledgement**

This work was supported by the Ministry of Science and Technology, Taiwan under the Project Number MOST 107-2811-
M-008-033 and Taiwan Environmental Protection Administration under Project Number 107D081.The authors gratefully
acknowledge all assistants involved in the system installation, maintenance, and site operation at Mt. Lulin and Doi Ang
Khang stations. The 7-SEAS, MPLNET, and AERONET projects were supported by the NASA Earth Observing System and
Radiation Sciences Program. The authors would like to acknowledge EPA Taiwan, CWB Taiwan and PCD Thailand for the
provision of ground-based measurement dataset, as well as MODIS and CALIPSO for satellite products/imagery.

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
