# Peer review of "Improving prediction of trans-boundary biomass burning plume dispersion: from northern peninsular Southeast Asia to downwind western north Pacific Ocean"

_Atmospheric Chemistry and Physics, 2020_

## Referee Comment (RC1)

In this paper, different plume rise methods for biomass burning (BB) are tested according to their impact on the simulation of trans-regional transport of BB aerosols in CMAQ model. The results emphasize the importance of applying the in-line plume rise module which calculates the plume height based on real-time atmospheric stratification. In addition, the paper shows the empirically assigned daily variation of smoldering fraction and fire size also play a role. With the help of model simulations, the paper analyzes the ways in which BB smoke transports to Taiwan and interacts with the near-surface pollutants.

The results of this paper is valuable because it is important for future models to figure out how those in-line plume rise modules perform and then to improve them accordingly. Although several similar plume rise modules have been developed for many years, limited works were published to evaluate the effect of such modules by multiple observations like this one. However, the paper still needs further revision because of bad organization and presentation.

General comments:

1. The authors declare "three distinct transport mechanisms" could bring BB aerosol from nPSEA to Taiwan both in the Abstract and the Section 4. However, the three mechanisms all depends on the **same** mechanism (strong westerlies in the free Troposphere) to **transport** BB aerosol from nPSEA to Taiwan. The differences among the three mechanisms in fact lie on how the BB aerosol interacts with the local pollutants in Taiwan. Therefore, it might be inappropriate to say "three transport mechanisms" in the context of East Asian region. Please consider to change those expressions according to the descriptions in line 450 which in fact make more sense.

2. The authors decide to compare the simulation from 2013 with observations from 2014 in Section 3.2 because of "**incomplete** MPLNET dataset of 2013". However, the reason is not convincing enough. On one hand, figure 4a shows quite **complete** MPLNET data coverage in 2013 which seems enough to give an average of good quality. On the other hand, the authors could do a simulation of 2014 and then get similar results if fire conditions are similar between 2013 and 2014 (as stated in lines 248-250).

3. Section 3.3 is not well organized which makes the reader difficult to follow.

For example, the authors seem to indicate the high resolution of FINN inventory plays the key role in the good performance of the in-line calculation. However, without another experiment using a lower emission resolution and the same in-line calculation as a comparison, such indication is only a speculation and should not appear in the conclusion (lines 455-457) as a strong argument.

Also, "BB emission is mainly caused by small fires and dry conditions over the period in the region" is not enough for the readers to understand "why the inline module worked well to represent the BB condition". I guess the heights prescribed in the off-line module tend to overestimate the plume height under dry conditions (dryer atmospheric stratification damps the pyro-convection through entrainment). Therefore, the in-line module which considers the atmospheric condition performs better. Anyway, more information is needed.

At last, is Figure 8 represents the near surface level or some upper levels? In either case the

corresponding statement is needed in Figure 8. In addition, is it possible that the difference between Figure 8b and 8c results from the different smoldering fraction between FWrp and IWrp+EC. As shown in Figure 2c and 2e, at 17:00 LST (around 9:00 UTC), FWrp (IWrp+EC) happens to have small (big) smoldering fraction which means little (much) aerosol is emitted near the surface and fire hotspots are therefore unclear (clear) in figure 8b. If so, the authors should reconsider the validity of some statements in Section 3.3.

4. Finally not mandatory and only a suggestion, is it possible to add another simulation in which biomass burning pollutants are emitted directly into the first model layer (near surface layer)? Such setup is still used in many (even some of the most state-of-art) models and probably works better as a control or benchmark than the "Nofire" setup if the authors want to emphasize the importance of in-line plume rise module.

Comments by line:

22. It might be confusing to use "the calibrated model" because no specific **calibration** is mentioned in the abstract. Use "Such setup" or "This measures" might be better.

23. The authors might want to say "BB aerosol concentration prediction" instead of "BB emission prediction". Emission is the flux at which the pollution is emitted to the atmosphere and usually not predicted by the model. Please check the whole paper to avoid similar mistakes.

24. Please consider to remove the contents inside the brackets. It is unnecessary and even confusing to mention the observation type like MODIS AOD and CALIPSO in this way.

52. "the vertical distribution percentage of BB **emission** was".

65. "interaction" between what and what?

81. "supply" instead of "supplies". Please check the whole manuscript to avoid such grammatical mistakes.

Table 1. Please include information about the **emission inventory** for **D04**.

156. Only one "and" is needed.

170. "Burnt area size" is not the same as "Fire size". "Fire size" is more proper in this context.

Table 2. In line "IDef", "Smoldering fraction: **yes**" makes no sense. Please check if it is a mistake. Otherwise, more details are needed.

224. "The **systematic peaks** for these pollutants are believed to be the uncertainties involving the FINN BB emission". This sentence is confusing. Do the authors mean "the systematic error for"?

Table 3. "MNB" and "MNE" still exist without explanation. Please check the whole manuscript to replace them with "MFB" and "MFE", Also, it is strange to find R decreases when MNB and MNE both decrease from IWrp to IWrp+EC. More explanation might be needed.

Figure 3. The unit "ug/m-3" is wrong. Please use either ug/m3 or ug*m-3. Also, colors look different between lines in legend and figure. (For example, the line representing observation is black in the figure but appears to be gray in the legend).

240. Is the lidar "MPLNET v0 L1.5a" at the same position as DAK station? If so, please indicate it in section 2.1 and in Figure 1.

249. Please replace the letter "x" (**ex**) with symbol "×" (**multiply**). Also check the whole paper

to avoid similar mistakes.

362. The sentence "It is because the boundary layer height⋯⋯" is out of context. Please delete it or reorganize the context.

408. The sentence "The detection of BB **intrusion into surface** sites ⋯⋯" is not consistent with the context. The mechanism mainly describes the near surface pollutants get upward and mix with the BB smoke above, rather than BB smoke gets downward and intrude into surface.

416. "BB aerosols have the most direct influence on the surface site in western Taiwan" is not enough for readers to understand why it "is coherent to the reduction of surface O3, NOx, and SO42- aerosols in 2006". More explanation is needed.

422. Please reorganize the sentence "The allocation fraction will need to improve looking⋯⋯" which is difficult to understand.

Figure D2. The figure has never been mentioned in the main text. It could be removed if it is unnecessary for your conclusions.

---

## Referee Comment (RC2)

The manuscript conducted model sensitivity analyses to compare a few different plume rise approaches in WRF-CMAQ over the northern peninsular Southeast Asia (nPSEA), and used the best case to study the transport of biomass burning aerosol to Taiwan. While this is an interesting and important topic, the manuscript needs to address the following comments before publication.

1. Line 13: "The boreal spring biomass burning (BB) in the northern peninsular…". Change it to "plumes from the boreal spring biomass burning"or "trace gases and aerosols emitted from the boreal spring biomass burning".

2. Line 16: Provide full name for WRF-CMAQ in the abstract.

3. Line 23: "The calibrated model greatly improves not only the BB emission prediction over…" Do you mean "improves not only the prediction of BB impact"?

4. Lines 45-47: change "overpredict the BB emissions" and "exceedance of predicted emission" to "overestimate the BB emissions" and exceedance of estimated emission".

5. The temporal and spatial domain of the study is limited. It would be nice to at least include a discussion/implication of broad application outside the time and region of the study domain.

6. Line 83: add full name for ARW.

7. Line 85: "The model domain is dynamically nested down…". I'm not sure if I understand the term "dynamically nested". Please rephrase.

8. Line 88: Add a reference for the NCEP dataset.

9. Lines 97-104: This part describes observations used in this study, and is not part of sub-section "2.1 Model Physics and Experimental Design". I suggest make it a new sub-section 2.2.

10. Table 1: In the row "Emission inventory", also include the BB inventory FINNv1.5.

11. Since the study uses the version 1.5 of FINN (FINNv1.5), please make sure to use the term "FINNv1.5" instead of "FINN" (for example Line 139) in the text.

12. Figure 2. Please check if there's any error with Fig 2a and Fig 2b. It seems that F800 has a higher top than F2000.

13. Figure 3: The legend is not clear. For example, obs should be black instead of grey.

14. Lines 216-225: Please add more discussion on the reasons of the model biases in addition to the description of the figure details.

15. Table 3: Please add in caption why some numbers are bold while others are not.

16. Lines 246-259. I'm concerned with comparing the model results of 2013 with obs of 2014. The authors need to justify the reliability of such comparison. The fact that there are a similar number of burning hotspots in model domain 2 in 2013 and 2014 is far from enough. Even if the total number of fires are similar over model domain 2 in 2013 and 2014, their spatial distributions may be different. Meteorology may be different too. I do not think such comparison is valid unless the author further justify this. Alternatively, the author could run the model for 2014 and do the comparisons, or simply compare the obs and model results for 2013 (make sure to exclude model data when obs are not available for comparisons).

17. Lines 272-287: Figure 5 shows a very interesting case study with satellite data. However, it would be better to use model results to support some of the statements instead of using the empirical statements. For example, "The aerosol layers are believed to be lifted …", "It is known that the burning aerosols…".

18. Lines 284-285: "Recently, it is proven through brute-force methods that the pollution from clusters arrived at the higher altitude in Taiwan during the winter season.". I'm not sure I understand this sentence. Please add more details/explanations (for example, what clusters).

19. Lines 297-298: By "Figure 5", do you mean "Figure 6"?

20. Line 314: "The cross-sectional profile in Fig. 6 shows that the amount of emission produced by the offline method is substantially larger". For the simulations with fires in this study, emissions should be produced by FINNv1.5, instead of the offline method.

21. Line 314-317: "The cross-sectional profile in Fig. 6 shows that the amount of emission produced by the offline method is substantially larger than the amount produced by the inline method. Therefore, the total columnar AOD data provided by 1º x 1º MODIS Terra Level 3 AOD product (MOD08_D3, Platnick et al, 2015) during the same period (20 Mar 10:30 LST) is used for the verification of the aerosol concentration." I don't see the connection here. Please explain why "total columnar AOD data provided by 1º x 1º MODIS Terra Level 3 AOD is used" because "the amount of emission produced by the offline method is substantially larger than the amount produced by the inline method".

22. Figure 8: Please add in the Figure caption which model layer/level is shown.

23: Line 384: "This is the commonly known scenario that is well studied due to the availability 385 of measurement collected at LABS." Add a reference here.

24: 393: Change "The interaction of BB with local pollutants" to "The interaction of BB plumes with local pollutants".

25: Some of the statements/conclusions made in the manuscript are not supported by the analysis/figures/tables of the manuscript. There seems to be a mix of data analysis and literature review. For example, in the Conclusion, "impact on surface sites in Taiwan" is mentioned. However, the paper does not provide analysis for surface sites in Taiwan.

26: The connection between domain 03 and domain 04 needs to be further justified. While Figure 3 shows that the enhanced pollutants in some period is due to BB, it is not convincing enough that pollutants observed at LABS they are due to BB in domain 03.

---

## Author Comment (AC1)

**Reviewer report 1:**

**General comments:**

**Comment #1**: The authors declare "three distinct transport mechanisms" could bring BB aerosol from nPSEA to Taiwan both in the Abstract and the Section 4. However, the three mechanisms all depends on the same mechanism (strong westerlies in the free Troposphere) to transport BB aerosol from nPSEA to Taiwan. The differences among the three mechanisms in fact lie on how the BB aerosol interacts with the local pollutants in Taiwan. Therefore, it might be inappropriate to say "three transport mechanisms" in the context of East Asian region. Please consider to change those expressions according to the descriptions in line 450 which in fact make more sense.

**Answer #1:** Agree. Sentence revised as below.

**Revision #1**: The BB aerosols from nPSEA is carried by the subtropical westerlies in free troposphere to the western North Pacific, while BB aerosol has found to interacts with the local pollutants in Taiwan region through three conditions: (a) overpass western Taiwan and enter central mountain area, (b) mix down to western Taiwan, (c) transport of local pollutants up and mix with BB plume on higher ground. The second condition that involves the prevailing high-pressure system from Asian cold surge is able to impact the most population in Taiwan.

**Comment #2**. The authors decide to compare the simulation from 2013 with observations from 2014 in Section 3.2 because of "incomplete MPLNET dataset of 2013". However, the reason is not convincing enough. On one hand, figure 4a shows quite complete MPLNET data coverage in 2013 which seems enough to give an average of good quality. On the other hand, the authors could do a simulation of 2014 and then get similar results if fire conditions are similar between 2013 and 2014 (as stated in lines 248-250).

**Answer #2:** After going through previous literature for year 2013 in Pani et al (2016) and year 2014 in Wang et al (2015). We agree that the aerosol extinction profiles are indeed different, hence using data from year 2014 to represent for year 2013 is not sensible. Hence, we have extracted data from year 2013 from MPLNET for the subsequent comparison.
*Pani, S. K., Wang, S. H., Lin, N. H., et al.: Radiative effect of springtime biomass-burning aerosols over northern indochina during 7-SEAS/BASELInE 2013 campaign, Aerosol Air Qual. Res., 16(11), 2802–2817, https://doi.org/10.4209/aaqr.2016.03.0130, 2016.*
*Wang, S.-H., Welton, E. J., Holben, B. N., et al.: Vertical Distribution and Columnar Optical Properties of Springtime Biomass-Burning Aerosols over Northern Indochina during 2014 7-SEAS Campaign, Aerosol Air Qual. Res., 15, 2037–2050, https://doi.org/10.4209/aaqr.2015.05.0310, 2015a.*

**Revision. #2**: The 3-hourly average profile of the extinction coefficient from MPLNET v0 L1.5a, **IWrp+EC** and **FWrp** model output during 13 – 28 Mar 2013 at DAK station is illustrated in Fig. 4b-d. In Fig.4b, the MPLNET extinction coefficient is low at the surface and peaks between 2.5–3.2 km. The model output has a lower elevation over DAK station has modelled a higher extinction coefficient, which is likely to be accumulation effect due to lower wind condition. (Please refer to the manuscript for more write-ups)

[Figure]

Figure 4: Vertical extinction coefficient profiles between 13 to 28 Mar 2013 at DAK station from (a) MPLNET with boundary layer height (white), (b) MPLNET 3-hourly average extinction coefficient, (c) IWrp+EC 3-hourly averaged model output, (d) FWrp 3-hourly averaged model output.

**Comment #3a.**

Section 3.3 is not well organized which makes the reader difficult to follow.

For example, the authors seem to indicate the high resolution of FINN inventory plays the key role in the good performance of the in-line calculation. However, without another experiment using a lower emission resolution and the same in-line calculation as a comparison, such indication is only a speculation and should not appear in the conclusion (lines 455-457) as a strong argument.

**Answer #3a:** Agree. The argument is based on the understanding of each emission inventory and indeed without additional comparison run, this statement is not conclusive. The corresponding sentences are revised.

**Revision #3a (Section 3.3): The FINN dataset provides high-resolution data for each fire (1 km$^2$) compared to the other emission dataset (GFEDv4s: 0.25⁰; GFASv1.2: 0.1⁰). As the finest study domain at the burning source is downscaled to 5km, the FINN dataset would have the nearest representation of the emission grid distribution. BB emission in the nPSEA is mainly caused by small fires and dry conditions over the period (Giglio et al., 2013; Reid et al., 2013), hence the representation of the small fires (usually accounted from 500 m burnt area) in the emission inventory is crucial. This might have been one of the reason that it fits better in.** the inline calculation with the plume-in-grid concept.

**Revision #3a (Conclusion)**: The sub-grid scale allocation of the BB emission requires **fitting and testing of** BB emission inventory **to make sure it reproduces** the individual fires with distinct and realistic peaks.

**Comment #3b**: Also, "BB emission is mainly caused by small fires and dry conditions over the period in the region" is not enough for the readers to understand "why the inline module worked well to represent the BB condition". I guess the heights prescribed in the off-line module tend to overestimate the plume height under dry conditions (dryer atmospheric stratification damps the pyro-convection through entrainment). Therefore, the in-line module which considers the atmospheric condition performs better. Anyway, more information is needed.

**Answer #3b:** Agree that the explanation is incomplete. The statements are being revised for clarity.

**Revision #3b (Section 3.3)**: **From this study, it is seen that the prescribed heights in the offline method have overestimated the plume rise height under the dry weather condition where the atmospheric stratification has no control on the pyro-convection through entrainment. While, the inline module (IWrp+EC) considers the variability of atmospheric condition over the mountain region better**.

**Comment #3c:** At last, does Figure 8 represent the near surface level or some upper levels? In either case the corresponding statement is needed in Figure 8.

**Answer #3c:** Figure 8 represents the near surface level. The caption is revised to clarify the figure content.

**Comment #3d:** In addition, is it possible that the difference between Figure 8b and 8c results from the different smoldering fraction between FWrp and IWrp+EC. As shown in Figure 2c and 2e, at 17:00 LST (around 9:00 UTC), FWrp (IWrp+EC) happens to have small (big) smoldering fraction which means little (much) aerosol is emitted near the surface and fire hotspots are therefore unclear (clear) in figure 8b. If so, the authors should reconsider the validity of some statements in Section 3.3.

**Answer #3d:** Additional run (F0: offline at near surface only) as suggested in Comment #4 is included to look at the role of the smoldering fraction on the near surface distribution of $PM_{10}$. For the F0 scenario, no smouldering is included, but it still has a higher near surface $PM_{10}$ concentration compared to FWrp with smouldering, and similarly, there is no distinguishable "hotspot" of $PM_{10}$ seen in both F0 and FWrp. Hence, we understood that the "hotspot" is not due to the role of smouldering but the role of the inline plume rise instead.

[Figure]

Figure 8: Spatial distribution of near surface PM10 concentration on 19 Mar 17:00 LST over burning regions of nPSEA for 4th domain (d04)

**Comment #4.** Finally not mandatory and only a suggestion, is it possible to add another simulation in which biomass burning pollutants are emitted directly into the first model layer (near surface layer)? Such setup is still used in many (even some of the most state-of-art) models and probably works better as a control or benchmark than the "Nofire" setup if the authors want to emphasize the importance of in-line plume rise module.

**Answer #4:** Thanks for the suggestion, we have included the additional simulation which the biomass burning pollutants is injected to the first model layer ("F0") for the sensitivity analysis in Section 3.1. However, for the subsequent part in Section 3, the result has shown that the offline module "FWrp" has higher accuracy compared to "F0", hence we will retain the current setting. In Section 4 where transport of biomass burning aerosol to Taiwan is concerned, we have decided to remain with the "Nofire" since the comparison is trying to differentiate the source of the aerosol whether it is local emission or trans-boundary biomass burning aerosols, but not focus on testing of the inline plume rise modules. In Fig. 7, the F0 scenario is similar to FWrp. Due to the near surface distribution of BB emission in F0, the main difference in the lower AOD present over the sea, where the emission from F0 is very likely to have deposited. In Fig.8, the near surface $PM_{10}$ concentration is compared and the distribution profile of F0 is again similar to FWrp, but F0 has expectedly higher concentration due to the intial distribution of BB over near surface layer only.

[Figure]

Figure 7: Comparison of daily total column AOD on 20 Mar (10:30 LST) of model output (a) IWrp+EC, (b) FWrp, (c) Nofire, (e) F0 with (d) MODIS data from Figure 5. Vector profiles given in (a-c) are the surface wind profile.

[Figure]

Figure 8: Spatial distribution of near surface PM10 concentration on 19 Mar 17:00 LST over burning regions of nPSEA for 4th domain (d04)

**Revision #4 (Table 2)**: Case setup to evaluate PLMRIM performance

[revised manuscript text omitted]

**Comments by line:**

**Comment L22**. It might be confusing to use "the calibrated model" because no specific calibration is mentioned in the abstract. Use "Such setup" or "This measures" might be better.

**Answer:** Ok.

**Revised**: **Such setup** greatly improves not only the BB aerosol concentration prediction over near-source and receptor ground-based measurement sites but also the aerosol vertical distribution and column aerosol optical depth of the BB aerosol along the transport route.

**Comment L23**. The authors might want to say "BB aerosol concentration prediction" instead of "BB emission prediction". Emission is the flux at which the pollution is emitted to the atmosphere and usually not predicted by the model. Please check the whole paper to avoid similar mistakes.

**Answer:** Ok.

**Revised**: Such setup greatly improves not only the BB **aerosol concentration** prediction over near-source and receptor ground-based measurement sites but also the aerosol vertical distribution and column aerosol optical depth of the BB aerosol along the transport route.

**Comment L24.** Please consider to remove the contents inside the brackets. It is unnecessary and even confusing to mention the observation type like MODIS AOD and CALIPSO in this way.

**Answer: Ok.**

**Revised:** Such setup greatly improves not only the BB aerosol concentration prediction over near-source and receptor ground-based measurement sites but also the **aerosol vertical distribution and column aerosol optical depth** of the BB aerosol along the transport route.

**Comment L52.** "the vertical distribution percentage of BB emission was".

**Answer: Ok.**

**Revised:** In those models, the vertical distribution percentage of **BB emission** was set to be constant throughout the case.

**Comment L65.** "interaction" between what and what?

**Answer: Ok.**

**Revised:** Knowing that the atmospheric circulation over nPSEA is also affected by terrain, the work now intends to incorporate **the interaction of the atmospheric stratification and BB plumes** into the PLMRIM.

**Comment L81.** "supply" instead of "supplies". Please check the whole manuscript to avoid such grammatical mistakes.

**Answer: Ok.**

**Revised:** The 7-SEAS spring campaigns carried out during the BB season **supply** abundance of data to the near source burning and receptor.

**Comment Table1**. Please include information about the emission inventory for D04.

**Answer: Ok.**

**Revised (Table1):** d01, d02, **d04**: MICS-ASIA 2010, biogenic emission from MEGANv2.1

**Comment L156.** Only one "and" is needed.

**Answer: Ok.**

**Revised:** The inline plume rise algorithm couples the interaction of BB plumes dispersion with the basic weather dynamics to determine the effective plume rise height**, subsequently** the plume top and bottom.

**Comment L170.** "Burnt area size" is not the same as "Fire size". "Fire size" is more proper in this context. Table 2. In line "IDef", "Smoldering fraction: yes" makes no sense. Please check if it is a mistake. Otherwise, more details are needed.

**Answer: Ok.**

**Revised:** IWrp has updated Idef with the WRAP empirical specification on **fire size**.

**Comment L224**. "The systematic peaks for these pollutants are believed to be the uncertainties involving the FINN BB emission". This sentence is confusing. Do the authors mean "the systematic error for"?

**Answer: Ok.**

**Revised:** The **systematic errors for these pollutants at the peak points** are believed to be the uncertainties involving the FINN BB emission (Pimonsree et al., 2018).

**Comment Table 3**. "MNB" and "MNE" still exist without explanation. Please check the whole manuscript to replace them with "MFB" and "MFE", Also, it is strange to find R decreases when MNB and MNE both decrease from IWrp to IWrp+EC. More explanation might be needed.

**Answer:** The explanation for MNB and MNE are included at the text, table caption and Appendix C. Result is cross-checked to make sure that it is correct. While comparing IWrp to IWrp+EC, the reduction of MNE and MNB indicates the error from the model data has converged towards the observation, in other words, the value from simulation has become closer to observation. While reduction of R is the weaken correlation for the entire simulation and observation dataset, in other words, the overall trend of between two dataset might have changed. Hence, these two statistical indicators are different in terms of interpretation of the result accuracy, and it is not impossible to have the reduction of R with the reduction of the error bias. In this study, the improvements of result shown by the reductions of MNE and MNE have a strong signals, up to 50% improvements for MFE of daily $PM_{10}$ and MNE of hourly $O_3$ and CO. The reduction of R is approximately 1% for daily $PM_{10}$ and hourly CO, which have been rather insignificant and might possibly be the numerical noise. Hence, the overall performance of IWrp+EC is still to be considered as improved in this context.

**Revised (Table 3)**: Table 3 shows the performance of PLMRIM on daily $PM_{10}$, daily $PM_{2.5}$, hourly $O_3$ and hourly CO at LABS and DAK according to the model benchmark (correlation coefficient, R; Mean Fractional Bias, MFB; Mean Fractional Error, MFE; **Mean Normalized Bias, MNB; Mean Normalized Error, MNE**) suggested by the Taiwan EPA (Appendix C).

**Revised (Table 3 Caption)**: Performance of modelled chemistry field with different settings of PLMRIM at mountain site in western North Pacific (LABS) and nPSEA (DAK). R: correlation

coefficient; MFB: Mean Fractional Bias; MFE: Mean Fractional Error; **MNB: Mean Normalized Bias; MNE: Mean Normalized Error.**

**Revised (Appendix C):**

$$\text{Mean Normalized Bias (MNB): MNB} = \frac{1}{N}\sum_{i=1}^{N}\left(\frac{M_i - O_i}{O_i}\right) \times 100\%$$

$$\text{Mean Normalized Error (MNE): MNE} = \frac{1}{N}\sum_{i=1}^{N}\left|\frac{M_i - O_i}{O_i}\right| \times 100\%$$

**Comment Figure 3**. The unit "ug/m-3" is wrong. Please use either ug/m3 or ug*m-3. Also, colors look different between lines in legend and figure. (For example, the line representing observation is black in the figure but appears to be gray in the legend).

**Answer:** Ok. The legend colour and label for figure unit are updated.

**Revised (Figure 3):**

[Figure]

Figure 3: Comparison of PLMRIM (observation (black), nofire (blue), FWrp (green), IDef (orange), IWrp+EC (red) of (a) hourly wind field and PM$_{2.5}$ at DAK, and (b,c,d) hourly wind field and (b) PM$_{10}$ (b), (c) CO, (d) O$_3$ at LABS in Mar 2013; Grey shade highlights the high pollution hour at LABS (CO > 300 ppb, PM$_{10}$ > 35 µg m$^{-3}$). Wind field for observation (black) and simulation (red) are shown in vector form.

**Comment L240**. Is the lidar "MPLNET v0 L1.5a" at the same position as DAK station? If so, please indicate it in section 2.1 and in Figure 1.

**Answer:** Ok.

**Revised**: In line with the 2014 7-SEAS spring campaign conducted in nPSEA, **the MPLNET device is located at the Doi Ang Khang Meteorology (DAK) Station to collect the near-source aerosol vertical distribution profile (v0 L1.5a) data.** The gridded extinction, diagnosed from the planetary

boundary layer height and vertical aerosol extinction coefficient data collected is used to verify the performance of the model output (Wang et al., 2015a).

**Revised (Figure 1 Caption):** Figure 1: (a) Domain setup of model (domain 1-4) with terrain height information; (b) 3$^{rd}$ domain covering Taiwan (d03) with information of terrain height (contour fill), AA' cross section (dotted red line), locations of Taiwan EPA air quality and CWB weather stations (black dots) and LABS receptor site (big red dot); (c) 4$^{th}$ domain covering part of nPSEA (d04) with terrain height (contour fill), BB' cross section (dotted red line), location of Thailand PCD ground air quality stations (black dots) and DAK source site (big red dot). **The latter also stationed the MPLNET device.**

**Comment L249.** Please replace the letter "x" (ex) with symbol "×" (multiply). Also check the whole paper to avoid similar mistakes.

**Answer**: Ok.

**Revised**: The incomplete MPLNET dataset of 2013 has prompted the use of the data from 2014 (Version 2 and Level 1.5) (Wang et al., 2015a) with a similar number of burning hotspots (sum of hotspot covered in model domain 2: 2013 = 1.1 $\times$ 10$^5$, 2014 = 1.2 $\times$ 10$^5$) and AOD (averaged from MERRA-2 AOD product in model domain 2: 2013 = 0.34, 2014 = 0.38) during the period of study.

**Revised**: Therefore, the total columnar AOD data provided by 1º $\times$ 1º MODIS Terra Level 3 AOD product (MOD08_D3, Platnick et al, 2015) during the same period (20 Mar 10:30 LST) is used for the verification of the aerosol concentration.

**Revised**: The boundary condition data in WRF model uses the reanalysis weather data. These data are assimilated with measurement data, they are available in coarse resolution (1° $\times$ 1°).

**Revised:** Given that the 3$^{rd}$ domain is of 5 km $\times$ 5 km resolution, the height of Mt. Lulin might be averaged out by the lower terrain surrounding it and the model height of Mt. Lulin is lower (2216 m, layer = 1) than its original height (2862 m).

**Comment L362.** The sentence "It is because the boundary layer height......" is out of context. Please delete it or reorganize the context.

**Answer:** Noted. The sentence is deleted.

**Comment L408**. The sentence "The detection of BB intrusion into surface sites ......" is not consistent with the context. The mechanism mainly describes the near surface pollutants get upward and mix with the BB smoke above, rather than BB smoke gets downward and intrude into surface.

**Answer**: Agreed. The sentence is better suited in Section 4.2 and hence moved over.

**Revised**: A larger amount of fine nanoparticles from local sources is measured at LABS especially during the morning even not during the **spring** burning season.

**Comment L416**. "BB aerosols have the most direct influence on the surface site in western Taiwan" is not enough for readers to understand why it "is coherent to the reduction of surface $O_3$, $NO_x$, and $SO_4^{2-}$ aerosols in 2006". More explanation is needed.

**Answer**: Agree on the confusion caused. Statement rephrased for clarity.

**Revised**: Among the three mechanisms, the BB aerosols have **a more** direct influence on the surface site in western Taiwan **under the second mechanism. Such condition occurred due to Asian continental cold surge that the high-pressure system moves south-eastwards. Under favourable upwind weather condition, the dust can be lifted and transported downwind to react with the BB aerosols. Such situation is shown on the co-existence of two major pollution event (dust and BB) that reduces the** surface $O_3$, $NO_x$, and $SO_4^{2-}$ aerosols over **western Taiwan** in 2006 (Dong et al., 2018).

**Comment L422**. Please reorganize the sentence "The allocation fraction will need to improve looking......" which is difficult to understand.

**Revised**: The allocation **of smoldering** fraction **in SEA** will need **to be improved to account of the tendency of** small fires to smolder (Akingunola et al., 2018; Zhou et al., 2018).

**Comment Figure D2**. The figure has never been mentioned in the main text. It could be removed if it is unnecessary for your conclusions.

**Answer:** Thanks for noticing. It's been removed.

---

## Author Comment (AC2)

**Reviewer report 2:**

**General comments:**

The manuscript conducted model sensitivity analyses to compare a few different plume rise approaches in WRF-CMAQ over the northern peninsular Southeast Asia (nPSEA), and used the best case to study the transport of biomass burning aerosol to Taiwan. While this is an interesting and important topic, the manuscript needs to address the following comments before publication.

Specific comments:

1. Line 13: "The boreal spring biomass burning (BB) in the northern peninsular...". Change it to "plumes from the boreal spring biomass burning" or "trace gases and aerosols emitted from the boreal spring biomass burning".

**Revision #1:** Changes "The boreal spring biomass burning" to "Plumes from the boral spring biomass burning"

2. Line 16: Provide full name for WRF-CMAQ in the abstract.

**Revision #2:** Included the full name for WRF-CMAQ in the abstract – "Weather Research and Forecast coupled with Community Multiscale for Air Quality model"

3. Line 23: "The calibrated model greatly improves not only the BB emission prediction over..." Do you mean "improves not only the prediction of BB impact"?

**Answer #3:** Sorry for the confusion. This sentence is intended to mean the improvement of the BB aerosol concentration prediction.

**Revision #3: Such setup** greatly improves not only the BB **aerosol concentration** prediction over near-source and receptor ground-based measurement sites but also the aerosol vertical distribution and column aerosol optical depth of the BB aerosol along the transport route."

4. Lines 45-47: change "overpredict the BB emissions" and "exceedance of predicted emission" to "overestimate the BB emissions" and exceedance of estimated emission".

**Revision #4:** Previous studies have found that the numerical model has prone to **overestimate** the BB emissions including CO, $PM_{2.5}$, and $PM_{10}$ up to three times of the measured amount at the major burning source in northern Thailand (Huang et al., 2013; Pimonsree et al., 2018). The exceedance of **estimated** emission at the near-source burning leads to the incorrect modelled signal at the downwind site (Fu et al., 2012).

5. The temporal and spatial domain of the study is limited. It would be nice to at least include a discussion/implication of broad application outside the time and region of the study domain.

**Answer #5:** Additional discussion and implication have been included in Section 3.3

[revised manuscript text omitted]

9. Lines 97-104: This part describes observations used in this study, and is not part of sub-section "2.1 Model Physics and Experimental Design". I suggest make it a new sub-section 2.2.

**Answer #9:** Apology for the confusion. This part is intended as the data used for model verification which is a continual from the discussion before. Additional description is included to ensure the flow of the paragraphs.

**Revision #9: On top of the ground-based measurement weather and air quality data, the lidar systems are also used to evaluate the performance of the model ability to estimate the vertical profile of BB aerosols. They are the bottom-up Micro-Pulse Lidar Network (MPLNET) and top-down Cloud-Aerosol Lidar with Orthogonal Polarization (CALIOP) lidar sensors.** The MPLNET is a federated network managed by NASA to measure the aerosol vertical structure (Welton et al., 2000). In line with the 2014 7-SEAS spring campaign conducted in nPSEA, **the MPLNET is located at the Doi Ang Khang Meteorology (DAK) Station to collect the near-source aerosol vertical distribution profile (L1.5a) data.** The gridded extinction, diagnosed from the planetary boundary layer height and vertical aerosol extinction coefficient data collected is used to verify the performance of the model output (Wang et al., 2015a). The CALIOP **sensor mounted** on the Cloud-Aerosol Lidar and Infrared Pathfinder Satellite Observations (CALIPSO) satellite is used to study the transport pattern over larger spatial coverage to complement the single point cross-extinction profile provided by the MPLNET system. The diagnosed vertical feature mask (VFM) product is used to distinguish the aerosol types with consideration of observed backscatter strength and depolarization (Winker et al., 2011).

10. Table 1: In the row "Emission inventory", also include the BB inventory FINNv1.5.

**Revision #10:**

**Table 1: WRF and CMAQ model settings**

|  | **Settings** |
|---|---|
| **Weather model** | WRF version 3.9.1 |
| **Period** | 1– 31 Mar 2013 (after spin up) |
| **Boundary condition** | NCEP FNL lateral boundary condition |
| **Vertical** | 41 layers up to 50 hPa with 10 layers in the bottom 2km |
| **Weather nudging** | Grid and observation nudging |
| **Planetary boundary** | Asymmetric Convective Mechanism 2 |
| **Surface and land surface model** | Pleim-Xiu |
| **Longwave radiation** | RRTM scheme |
| **Shortwave radiation** | Goddard |
| **Microphysics scheme** | Goddard |
| **Cumulus scheme** | Kain-Fritsch (1) for d01, d02 only |
| **Chemistry transport model** | CMAQ version 5.2.1 |
| **Gas-phase chemistry and aerosol mechanism** | CB05e51 + AE6 (with aqueous chemistry) |
| **Anthropogenic and biogenic emission inventory** | d01, d02, d04: MICS-ASIA 2010, biogenic emission from MEGANv2.1
d03: Taiwan local emission inventory (TEDS v8.1) |

11. Since the study uses the version 1.5 of FINN (FINNv1.5), please make sure to use the term "FINNv1.5" instead of "FINN" (for example Line 139) in the text.

**Revision #11:** The global data set, Fire INventory from NCAR (FINN v1.5**, referred as "FINN" here onwards**) has been applied in several previous works of literature in the region (Lin et al., 2014; Pimonsree and Vongruang, 2018) and is used as the input to the BB emission inventory into the model.

12. Figure 2. Please check if there's any error with Fig 2a and Fig 2b. It seems that F800 has a higher top than F2000.

**Revision #12: (Figure 2)**: Initial CO emission rate (mol/s) profile at Mae Hong Son, Thailand on 13 Mar 2013 (UTC) for each case setup in Table 2 with (a) F0, (b) F800, (c) F2000, (d) FWrp, (e) IDef, (f) IWrp/IWrp+EC.

[Figure]

13. Figure 3: The legend is not clear. For example, obs should be black instead of grey.

**Revision #13:** Figure 3

[Figure]

Figure 3: Comparison of PLMRIM (observation (black), nofire (blue), FWrp (green), IDef (orange), IWrp+EC (red) of (a) hourly wind field and PM$_{2.5}$ at DAK, and (b,c,d) hourly wind field and (b) PM$_{10}$ (b), (c) CO, (d) O$_3$ at LABS in Mar 2013; Grey shade highlights the high pollution hour at LABS (CO > 300 ppb, PM$_{10}$ > 35 µg m$^{-3}$). Wind field for observation (black) and simulation (red) are shown in vector form.

14. Lines 216-225: Please add more discussion on the reasons of the model biases in addition to the description of the figure details.

**Answer #14:** The statistical indexes of the ground-based measurement have been discussed thoroughly in the beginning of Section 3.1 to supplement the discussion of time-series. Please see the Section 3.1 for more details.

15. Table 3: Please add in caption why some numbers are bold while others are not.

**Revision #15:** Table 3 caption - Performance of modelled chemistry field with different settings of PLMRIM at mountain site in western North Pacific (LABS) and nPSEA (DAK). R: correlation

coefficient; MFB: Mean Fractional Bias; MFE: Mean Fractional Error; MNB: Mean Normalized Bias; MNE: Mean Normalized Error. **Bold values are model output that satisfied the standard of each index.**

16. Lines 246-259. I'm concerned with comparing the model results of 2013 with obs of 2014. The authors need to justify the reliability of such comparison. The fact that there are a similar number of burning hotspots in model domain 2 in 2013 and 2014 is far from enough. Even if the total number of fires are similar over model domain 2 in 2013 and 2014, their spatial distributions may be different. Meteorology may be different too. I do not think such comparison is valid unless the author further justify this. Alternatively, the author could run the model for 2014 and do the comparisons, or simply compare the obs and model results for 2013 (make sure to exclude model data when obs are not available for comparisons).

**Answer #16:** After going through previous literature for year 2013 in Pani et al (2016) and year 2014 in Wang et al (2015). We agree that the aerosol extinction profiles are indeed different, hence using data from year 2014 to represent for year 2013 is not sensible. Hence, we have extracted data from year 2013 from MPLNET for the subsequent comparison.
*Pani, S. K., Wang, S. H., Lin, N. H., et al.: Radiative effect of springtime biomass-burning aerosols over northern indochina during 7-SEAS/BASELInE 2013 campaign, Aerosol Air Qual. Res., 16(11), 2802–2817, https://doi.org/10.4209/aaqr.2016.03.0130, 2016.*
*Wang, S.-H., Welton, E. J., Holben, B. N., et al.: Vertical Distribution and Columnar Optical Properties of Springtime Biomass-Burning Aerosols over Northern Indochina during 2014 7-SEAS Campaign, Aerosol Air Qual. Res., 15, 2037–2050, https://doi.org/10.4209/aaqr.2015.05.0310, 2015a.*

**Revision #16**: The 3-hourly average profile of the extinction coefficient from MPLNET v0 L1.5a, **IWrp+EC** and **FWrp** model output during 13 – 28 Mar 2013 at DAK station is illustrated in Fig. 4b-d. In Fig.4b, the MPLNET extinction coefficient is low at the surface and peaks between 2.5–3.2 km. The model output has a lower elevation over DAK station has modelled a higher extinction coefficient, which is likely to be accumulation effect due to lower wind condition. (Please refer to the manuscript for more write-ups)

[Figure]

Figure 4: Vertical extinction coefficient profiles between 13 to 28 Mar 2013 at DAK station from (a) MPLNET with boundary layer height (white), (b) MPLNET 3-hourly average extinction coefficient, (c) IWrp+EC 3-hourly averaged model output, (d) FWrp 3-hourly averaged model output.

17. Lines 272-287: Figure 5 shows a very interesting case study with satellite data. However, it would be better to use model results to support some of the statements instead of using the empirical statements. For example, "The aerosol layers are believed to be lifted ...", "It is known that the burning aerosols...".

**Answer #17:** Due to the coarse time intervals of the satellite data, it is difficult to provide affirmative statement at this point, however, the model result in subsequent section is able to confirm the observation from the satellite data. The sentence is rephrased to inform the author the subsequent section will prove the statements on satellite data.

**Revision #17:** The aerosol layers are believed to be lifted to a higher level and also mixed to the surface over the land mask in southeastern China, **which is later confirmed in the model result in Section 4.**

18. Lines 284-285: "Recently, it is proven through brute-force methods that the pollution from clusters arrived at the higher altitude in Taiwan during the winter season.". I'm not sure I understand this sentence. Please add more details/explanations (for example, what clusters).

**Revision #18:** Recently, it is proven through brute-force methods that the pollution from the **PRD** cluster arrived at the higher altitude in Taiwan during the winter season (Chuang et al., 2019).

19. Lines 297-298: By "Figure 5", do you mean "Figure 6"?

**Revision #19: Figure 6** shows the model $PM_{10}$ result for FWrp (range: 0-300 µg m-3) and IWrp+EC (range: 0-120 µg m-3) for the corresponding period of CALIPSO swath in Fig. 5.

**Answer #20:** The offline method is also using the data from FINNv1.5 to run. Please refer to Table 2 for detailed case setup.

**Answer #21**: The sentence is rephrased for clarity.

**Revision #21**: The cross-sectional profile **of PM$_{10}$** in Fig. 6 shows that the amount of emission produced by the offline method is substantially larger than the amount produced by the inline method. **However, it could not be verified the vertical PM$_{10}$ value due to the lack of measurement of vertical distribution of PM$_{10}$. The amount of PM$_{10}$ has directly contributed to the columnar AOD value and the latter could serve as a good benchmark for the accuracy of model aerosol concentration**. Hence, the total columnar AOD data provided by $1^{\circ} \times 1^{\circ}$ MODIS Terra Level 3 AOD product (MOD08_D3, Platnick et al, 2015) during the same period (20 Mar 10:30 LST) is used for the verification of the aerosol concentration **through the columnar AOD value**.

22. Figure 8: Please add in the Figure caption which model layer/level is shown.

**Revision #22**: Figure 8: Spatial distribution of **near surface** PM$_{10}$ concentration on 19 Mar 17:00 LST over burning regions of nPSEA for 4th domain (d04)

23: Line 384: "This is the commonly known scenario that is well studied due to the availability 385 of measurement collected at LABS." Add a reference here.

**Revision #23**: This is the commonly known scenario that is well studied due to the availability of measurement collected at LABS **(Lee et al., 2011; Ou-Yang et al., 2014)**.

*Lee, C. Te, Chuang, M. T., Lin, N. H., Wang, J. L., Sheu, G. R., Chang, S. C., Wang, S. H., Huang, H., Chen, H. W., Liu, Y. L., Weng, G. H., Lai, H. Y. and Hsu, S. P.: The enhancement of PM2.5 mass and water-soluble ions of biosmoke transported from Southeast Asia over the Mountain Lulin site in Taiwan, Atmos. Environ., 45(32), 5784–5794, https://doi.org/10.1016/j.atmosenv.2011.07.020, 2011.*

*Ou-Yang, C. F., Lin, N. H., Lin, C. C., Wang, S. H., Sheu, G. R., Lee, C. Te, Schnell, R. C., Lang, P. M., Kawasato, T. and Wang, J. L.: Characteristics of atmospheric carbon monoxide at a high-mountain background station in East Asia, Atmos. Environ., 89, 613–622, https://doi.org/10.1016/j.atmosenv.2014.02.060, 2014.*

24: 393: Change "The interaction of BB with local pollutants" to "The interaction of BB plumes with local pollutants".

**Revision #24**: The interaction of BB **plumes** with local pollutants depends on the loading of local pollutants present.

25: Some of the statements/conclusions made in the manuscript are not supported by the analysis/figures/tables of the manuscript. There seems to be a mix of data analysis and literature review. For example, in the Conclusion, "impact on surface sites in Taiwan" is mentioned. However, the paper does not provide analysis for surface sites in Taiwan.

**Answer #25:** The paper has provided thorough analysis for mechanism of biomass burning plumes to arrive at surface sites in Taiwan in Section 4.2: Mixing of BB emission with local pollution on surface.

26: The connection between domain 03 and domain 04 needs to be further justified. While Figure 3 shows that the enhanced pollutants in some period is due to BB, it is not convincing enough that pollutants observed at LABS they are due to BB in domain 03.

**Answer #26**: The 2$^{nd}$ domain (d02) that cover the transport route is used to show the connection between d03 and d04. The comparison of Fig. 7a and 7c is able to show the difference between fire and nofire cases which is solely contributed by the biomass burning plumes from nPSEA. Additional description for Fig. 7 is included to clarify for it. Besides that, the nofire case is designed to remove only the biomass burning emission within d02, hence the comparison between the fire and nofire case for receptor region (d04) is able to identify the source of the biomass burning plumes from the source (d03).

**Revision #26**: **Figure 7 shows the 2nd model domain (d02) that covers the transport route between the source (d04) and the receptor (d03) domains. The comparison between Fig. 7a and 7c is able to show the difference between fire and nofire cases which is solely contributed by the biomass burning plumes from nPSEA. The figure also** shows that the total column AOD produced by the inline module gives a closer approximation to the MODIS. FWrp greatly overestimates the aerosol produced by the BB emissions, while the inline module gives a closer agreement on northern Thailand and southern Vietnam.

---

## Referee Report (RR1)

I am satisfied with the replies to my previous comments. The authors obviously worked hard to do extra experiments and analysis which makes the manuscript in a much better shape now. However, some minor mistakes still exist and are outlined below. The paper could be published after properly correcting the mistakes.

Table 2: In line "IDef", the "Smoldering fraction" is said to be "yes" which makes no sense. Do the authors mean it is the same as FWrp? If so, please explicitly denote it in the table like the line "IWrp".

Line 340: Probably because I did not make myself clear enough and therefore the authors made a mistake here. Dryer environmental air once entrained into the plume will decrease its water vapor content and also the latent heat released during updraft. So, dry stratification favors weak pyro-convection and therefore low injection height. Then, it is better to say "where the atmospheric stratification **damps** the pyro-convection through entrainment" rather than "**has no control**". "**has no control**" would confuse readers to believe dry stratification could strengthen pyro-convection. Freitas et al. (2007) is recommended if the authors still find it hard to understand.

Line 427: "the dust can be lifted and transported downwind to **react** with the BB aerosols". I am not sure whether dust aerosols could have chemical reactions with BB aerosols. The authors might want to say the two kinds of aerosol react with NOx, O3 and SO2 gases.

Line 429: "NOx, and **SO42- aerosols** over western Taiwan in 2006 (Dong et al., 2018)". Please check Dong et al., 2018 again to make sure **SO42- aerosols** are discussed in this paper.

Freitas, S. R., Longo, K. M., Chatfield, R., Latham, D., Silva Dias, M., Andreae, M., et al. (2007). Including the sub-grid scale plume rise of vegetation fires in low resolution atmospheric transport models. *Atmospheric Chemistry and Physics, 7*(13), 3385-3398.

---

## Author Response (AR2)

Reviewer's comments:

**Comment #1:**
I am satisfied with the replies to my previous comments. The authors obviously worked hard to do extra experiments and analysis which makes the manuscript in a much better shape now. However, some minor mistakes still exist and are outlined below. The paper could be published after properly correcting the mistakes.

**Answer #1:**
The authors would like to sincerely thank the efforts and constructive comments and from the reviewer to improve the quality of this manuscript.

**Comment #2:**
Table 2: In line "IDef", the "Smoldering fraction" is said to be "yes" which makes no sense. Do the authors mean it is the same as FWrp? If so, please explicitly denote it in the table like the line "IWrp".

**Answer #2:**
Thanks for pointing out. The smoldering fraction used in IDef is indeed being developed through the WRAP algorithm. It has been updated to "FWrp" instead of "yes".

**Comment #3:**
Line 340: Probably because I did not make myself clear enough and therefore the authors made a mistake here. Dryer environmental air once entrained into the plume will decrease its water vapor content and also the latent heat released during updraft. So, dry stratification favors weak pyro-convection and therefore low injection height. Then, it is better to say "where the atmospheric stratification damps the pyro-convection through entrainment" rather than "has no control". "has no control" would confuse readers to believe dry stratification could strengthen pyro-convection. Freitas et al. (2007) is recommended if the authors still find it hard to understand.

**Answer #3:**
Thanks for identifying the unclear statement. The paper is very helpful to understand the different effect of dry and wet stratification on pyro-convection. The sentence is now corrected.

**Revision #3:**
From this study, it is seen that the prescribed heights in the offline method have overestimated the plume rise height under the dry weather condition where the atmospheric stratification damps the pyro-convection through entrainment.

**Comment #4:**
Line 427: "the dust can be lifted and transported downwind to react with the BB aerosols". I am not sure whether dust aerosols could have chemical reactions with BB aerosols. The authors might want to say the two kinds of aerosol react with NOx, O3 and SO2 gases.

**Answer #4:**
Agree with the reviewer's concern. There is still no finding suggesting the reaction of both aerosols, the sentence is revised for clarity.

**Revision #4:**
Under favourable upwind weather condition, the dust can be lifted and transported downwind and concurrently present with the BB aerosols.

**Comment #5:**
Line 429: "NOx, and SO42- aerosols over western Taiwan in 2006 (Dong et al., 2018)". Please check Dong et al., 2018 again to make sure SO42- aerosols are discussed in this paper.

**Answer #5:**
Thanks for identifying the typo. It is $SO_2$ instead of $SO_4^{2-}$.